

# Effect of double diffusion processes in the deep ocean on the distribution and dynamics of particulate and dissolved matter: a case study in Tyrrhenian Sea

Xavier DURRIEU DE MADRON[1]*, Paul BLIN[1], Mireille PUJO-PAY[2], Vincent TAILLANDIER[3], Pascal CONAN[2,4]

[1] CEFREM, CNRS-Université de Perpignan Via Domitia, Perpignan, France

[2] LOMIC, CNRS-Sorbonne Université, Banyuls/Mer, France

[3] LOV, CNRS-Sorbonne Université, Villefranche/Mer, France

[4] OSU STAMAR, CNRS-Sorbonne Université, Paris, France

*Correspondence to*: Xavir Durrieu de Madron (demadron@univ-perp.fr)

**Abstract**: This study examines CTD, ADCP and optical data collected during the PERLE-3 cruise in March 2020 between the surface and 2000 m depth over an east-west section of the Tyrrhenian Sea in the Mediterranean. The focus will be on the impact of double diffusion processes, in particular salt fingering, on the distribution and dynamics of particulate and dissolved matter. The staircases develop at the interface between the warm, salty Levantine Intermediate Water (LIW) and the colder, less salty Tyrrhenian Deep Water (TDW) in the centre of the basin with low hydrodynamic energy. The results show that thermohaline staircases formed by salt fingering significantly influence particle sedimentation and biogeochemical cycling in deep ocean environments by altering vertical flux patterns. These density steps create distinct vertical layers that act as physical barriers, slowing the descent of particles and facilitating their retention and aggregation. Retention of fine particles at density gradients promotes the formation of larger aggregates, affecting particle size distribution. The staircases also affect dissolved matter by creating pronounced concentration gradients of oxygen and nutrients, which can influence microbial activity and nutrient cycling.

**Keywords**: Mediterranean, Tyrrhenian Sea, Double diffusion, Salt fingering, Suspended particulate matter, Particle size spectra, Nitrate, Mineralization



## 1. Introduction


Gravitational settling of biologically derived particles, produced by marine plankton in
the surface layer, is the primary driver of vertical fluxes of material in the deep ocean (Newton
and Liss, 1990). In general, vertical mass fluxes of particles in the ocean decrease exponentially
with depth due to particle degradation and the increasing density of seawater (Omand et al.,
2020). Marine particles have a wide range of sizes, from micrometres (clays) to centimetres
(organic detritus). Particles generally tend to agglomerate to form aggregates of different sizes,
shapes, densities, and characteristics (Kiko et al., 2022). The change in geometry of the
aggregates as well as their excess density compared to seawater are the factors that determine
their sedimentation rate.
The water column in the ocean is composed by the superposition of water masses with
distinct thermohaline characteristics and often distant origins. Large density interfaces,
associated with strong vertical gradients in temperature (~0.25 °C/m) at seasonal thermoclines,
or salinity (~0.5 g/kg/m) at river plumes can occur in the upper ocean, resulting in the retention
of the less dense particles. In the deep ocean temperature and salinity gradients are much
weaker, and the effect of particle size becomes greater than the effect of density excess, so that
settling velocity generally increases with particle size. Thus settling velocities in the open ocean
is on the order of 5 x$10^{-4}$ mm/s to 1 mm/s for particles ranging from 2 µm to 300 µm (McCave,
49  1975).

However, in the deep ocean, in the transition zones between two water masses, areas of
enhanced density gradients can occur due to double diffusion processes, in particular those
associated with salt fingers. Salt fingering is a common process in the ocean (Radko, 2013). It
requires a stably stratified water column, where a layer of warm, salty water overlies a layer of
cooler, less salty water. Because molecular diffusion of temperature is about 100 times faster
than that of salinity, the salinity interface initially remains essentially unchanged as temperature
exchange occurs across it. As a result, the saltier water above this thin mixed layer of average
temperature becomes denser than the less salty water below. Such thermohaline diffusive
convection leads to salt finger instabilities at the interface: the saltier water sinks and the less
salty water rises. The mixing process is then repeated at the new interfaces, resulting in a
progressive thickening of the mixed layer and an increase in the vertical density gradients on
either side of the interface. If favourable conditions persist, step structures can develop and
reach thicknesses of several tens of meters. These staircases exhibit typical temperature and
salinity gradients on the order of 0.01 °C/m and 0.005 g/kg/m, respectively.
There is also the case of a layer of turbid water overlying clearer, denser water, such as
for hypopycnal particle-laden river plumes (e.g. Green, 1987; Hoyal et al., 1999; Hoshiba et al.,
2021). Instabilities may arise as a result of the different diffusivities of each of the constituents
(sediment, temperature, salinity) (Parsons and Garcia, 2000). Both double diffusive and settling
driven convection processes can drive fingering and mixing at the interface, and therefore,
double diffusive sedimentation can add up to the gravitational settling velocity and enhance the
scavenging of particles from the buoyant turbid plume. However, for large particles and dilute
concentrations, the flux is more dominated by gravitational settling and the double diffusive
sedimentation of suspended particles is considered negligible (e.g. Davarpanah Jazi and Wells,
73  2016).



Various experimental and numerical studies have explored the impact of density
interfaces on the sedimentation of particulate material, but to our knowledge very little research
has been applied to the deep oceanic environment. Laboratory experiments conducted for a
variety of individual particles geometries and densities (e.g. Prairie et al., 2013; Mrokowska,
2018; Doostmohammadi and Ardekani, 2014; Doostmohammadi and Ardekani, 2015; Verso et
al., 2019) generally show a decrease in settling velocity, as well as a reorientation of the
particles, during its passage through the transition layer formed by the density interface. The
initial decrease is followed by an increase, once the particle reaches the base of the interface,
but this settling velocity is always smaller in the lower layer due to its higher density. The initial
decrease in velocity in the interface is likely due to the entrainment of less dense water by the
particles and the drag of their wake as they cross the interface. Finally, Maggi (2013) shows
that sedimentation rates are very well correlated with size for mineral particles, but are much
less clear for biomineral or biological material. Thus, the sedimentation rate of a solid particle
alone will depend almost exclusively on its size. This is not the case for aggregates, where
composition, shape, excess density and porosity must also be taken into account. Kindler et al.
(2010) demonstrated that slowly sinking particles, like highly porous aggregates, can be
retained and therefore accumulate at density interfaces, increasing the likelihood of collisions
and subsequent aggregation. They suggested that this increase in retention time may affect
carbon transformation through increased microbial colonization and utilization of particles and
release of dissolved organics.
The Mediterranean Sea is a prime location for the observation and study of thermohaline
staircases due to its unique hydrographic conditions. The study presented here focuses on the
behaviour of particulate matter and dissolved elements in the Tyrrhenian Sea (Fig. 1), a region
known to be favourable for salt fingers with the formation of large staircases at the transition
between intermediate and deep waters (Durante et al., 2019). The warm and saline Levantine
Intermediate Water (LIW), formed in the eastern Mediterranean Basin, is found throughout the
Tyrrhenian Sea at depths between 200 and 700 m. Deeper is the Tyrrhenian Deep Water
(TDW), colder and less saline, formed by the mixing of the deep waters of the western
Mediterranean with the intermediate and deep waters of the eastern basin Falco et al., 2016).
Staircase structures, which extend over most of the central basin, are found at depths between
600 and 2,500 m and are tens or even hundreds of meters thick.
The dataset used in this work was obtained from the PERLE3 cruise (Pujo-Pay et al.,
2020) whose main objective is to study the formation of intermediate Levantine waters in the
eastern basin and their fate and transformation along their course in the Mediterranean Sea.
Here, we focused on a section between the Bay of Naples and southern Sardinia, which cuts the
Tyrrhenian Sea from east to west. The data collected included hydrological parameters
(temperature, salinity, density), hydrodynamic parameters (current velocity and direction), and
particulate (turbidity, large particle abundance) and dissolved (oxygen, nitrate) parameters in
the upper 2000 m of the water column. The two questions addressed here are (1) what are the
characteristics and development conditions of the notable staircase structures observed during
this cruise, and (2) what is the impact of these staircases on the distribution of dissolved and
particulate matter. The answers to these two questions will provide new insights into the
ecological consequences of these small-scale structures.

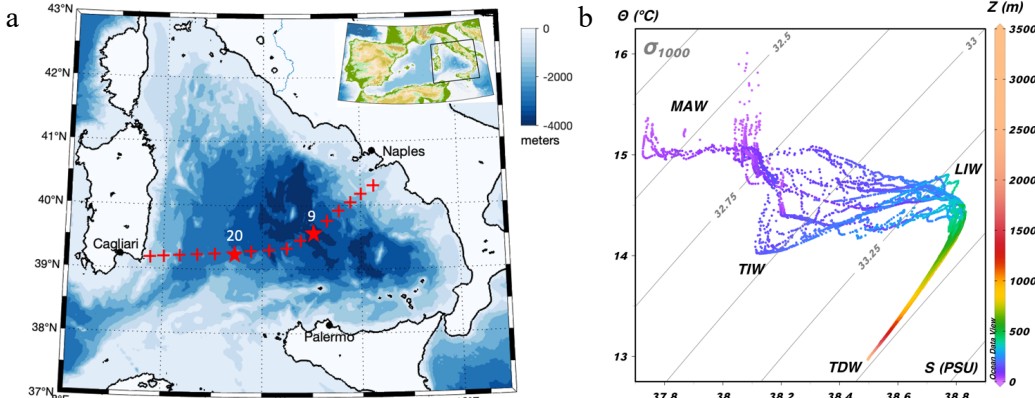

*Figure 1 (a) Map of the stations carried out during part of the PERLE-3 cruise (14–16 March 2020) in the Tyrrhenian Sea. Stations 9 and 20, indicated by stars, are the two stations chosen to characterize the water column respectively with and without staircases between intermediate and deep water. (b) θ-S diagram for the different stations of the transect and associated water masses (MAW, Modified Atlantic Water; TIW, Tyrrhenian Intermediate Water; LIW, Levantine Intermediate Water; TDW, Tyrrhenian Deep Water).*

## 2. Material and Methods

### 2.1 Shipborne CTD and optical data

This hydrographic survey of the Tyrrhenian Sea was conducted in March 2020 as part of the PERLE-3 cruise, which was unexpectedly shortened due to the outbreak of the COVID-19 pandemic. It includes 16 stations for which hydrological profiles were obtained between the surface and 2000 m with a Seabird 911+ CTD-O2 probe. Additional optical sensors (Wetlabs C-Star 0.25m pathlength transmissometer at 650 nm wavelength, Seabird Suna V2 UV nitrate sensor, Hydroptic Underwater Vision Profiler (UVP-5) were connected to the probe.

Data from the CTD sensors and the transmissometer were recorded at a frequency of 24 Hz. Data were therefore acquired every 2 cm at a descent rate of 1 m/s. The temperature and conductivity sensors provided measurements with a resolution of 0.0002 °C (4.10-5 S/m) and an accuracy of 0.002 °C (0,000 3 S/m), respectively. The dissolved oxygen sensor provided measurements with a resolution of 0.2 µmol/kg. The transmissometer has a resolution of 1.25 mV over a range of 0 to 5 V (WET Labs, Inc., 2011), giving a beam attenuation coefficient [BAC= — 4×ln (T%), where T% is the transmittance in %] resolution of $10^{-3}$ 1/m. The UVP-5 (Picheral et al., 2010) is a stand-alone instrument mounted on the CTD/rosette frame to quantify the vertical distribution of large particles and zooplankton. Images are acquired at a frequency of up to 6 Hz, i.e. on average every 20 cm at a descent rate of 1 m/s. The small size limit of the UVP-5 is determined by the optical resolution (94.7 µm corresponding to 1 pixel), while the large size limit is determined by the volume of water illuminated (1.02 l).





The stand-alone Seabird Suna V2 UV nitrate sensor attached to the CTD collected data
at 1 Hz from the surface down to 2000 m. It uses ultraviolet absorption spectroscopy to measure
nitrate in situ. A good correlation was observed with bottle measurements over the entire water
column collected during the cruise ($R^2$= 0.99, N=27, p <$10^{-5}$, $NO_{3\ SUNA}$ = 0.991 $NO_{3\ btl}$ − 0.116).
The CTD data processing was performed with the SBE Data Processing software, and
derived variables (conservative temperature, absolute salinity, potential density anomaly,
Brunt-Väisälä frequency and Turner angle) were estimated based on the TEOS-10 toolbox
(IOC, SCOR and IAPSO, 2010).
The transmissometer signal was associated with the finest and most numerous particles
(∼ 1–10 µm), while larger particles (from 80 µm) were seen by the UVP. For transmissometer
data, the turbidity spikes due to the passage of large particles through the beam were eliminated
by smoothing (Giering et al., 2020). For UVP data, the abundance (in # $l^{-1}$) of large particles
for the different size classes between 80 µm and 2000 µm (equivalent circular diameter) was
estimated from the raw images (Picheral et al., 2010). The UVP data were then processed to
estimate the particle size distribution, which was modelled using a typical power law of the
form $N(d) = C \times d^{-\alpha}$, where C represents a constant, d stands for particle diameter, and –denotes
the Junge index (Guidi et al., 2009). The Junge index is determined through linear regression
involving log-transformed values of N(d) and d. The calculation was performed on the size
range between 80 and 400 microns, in order to consider only the most abundant particles, which
represent to 94% of the total number observed and whose concentration is greater than 1 particle
per litre. The Junge index, $\alpha$, varies between 2.5 and 4.5 for this data set. A slope of the
differential particle size distribution of 4 indicates an equal amount of mass in logarithmically
increasing size intervals. Higher values indicate a greater dominance of fine particles within the
particle population, while lower values are associated with particle populations enriched in
larger particles.
The filtering step proved to be very important, given the scales on which the study
focused, to remove the spikes and reduce the noise of the signal while maintaining the best
vertical resolution. The 24 Hz raw data were subjected to outlier removal using two successive
moving median filters with window lengths of 7 and 5 scans, respectively. The data were then
binned at 1 m intervals and smoothed with a Loess regression filter. For temperature, salinity,
and potential density anomaly signals, the loess regression smoothing window length was 10
scans (10 metres). For dissolved oxygen, nitrate, beam attenuation coefficient, UVP, and Junge
index signals we applied a Loess regression smoothing with a window length of 50 scan (50
metres).
**2.2 Shipborne acoustical data for current and backscatter index**
During the PERLE-3 survey, two ship-borne Acoustic Doppler Current Profilers –
ADCPs – collected continuous current data from 21 m to 1200 m depth. The first ADCP was
an RDI OS150 with an acoustic frequency of 150 kHz, a sampling rate of 1 Hz, and a cell size
of 8 m, allowing a total range of 220 m. The second ADCP is an RDI OS38 with an acoustic
frequency of 38 kHz, a sampling rate of 1 Hz, and a cell size of 8 m, allowing a maximum range
of 1200 m. Data averaged over a 2-min period were concatenated and processed using
Cascade V7.2 processing software (Kermabon et al., 2018) to compute horizontal ocean current



velocities with a spatial resolution of 2 km, corrected for navigation and ship attitude
parameters, and filtered according to various quality criteria. Bathymetry (Etopo 1) was added
for bottom detection. The profile data for the meridional and zonal components of the current
for the two ADCPs were combined to obtain a complete profile between 21 and 1200 m depth
with maximum resolution in the surface layer.

In addition to the S-ADCP current measurements, current data between the surface and
2000 m depth were also collected using a dual-head L-ADCP (Lowered-Acoustic Doppler
Current Profiler) system. These measurements were collected with two RDI
Workhorse 300 kHz current meters mounted on the CTD frame, one looking up and one looking
down. The vertical profiling resolution was 8 m. The data were processed by the velocity
inversion method using version IX of the LDEO software (Thurnherr, 2021). Qualified external
data (CTD, S-ADCP, GPS) are used to process the L-ADCP data. Vertical ocean velocities
were calculated using the LADCP_w_ocean utility from combined raw L-ADCP and CTD data
(Thurnherr, 2022). The upward and downward looker data were processed separately and
combined during post-processing to provide vertical velocity profiles for the downcast and
upcast.

The data from the S-ADCP were also used to derive the acoustic backscatter index (BI,
Mullison, 2017), which a proxy for the abundance of centimetre-scale reflectors (organic
detritus, zooplankton, micronecton…) in the water column. This derivation takes into account
the absorption and geometric dispersion of sound $BI = K_c*(RL-Er) + (TL [w] + TL [g])$, where
$K_C$ is the conversion factor (*count to decibels*) of the ADCP used, RL the received signal, Er
the signal noise, TL [w] and TL [g] are the absorption and geometric dispersion factors of the
acoustic signal in water respectively.

**2.3 Profiling float CTD data**

Another CTD data set was collected from a BGC profiling float (WMO 6,902,903). This
float is equipped with SBE-41CP pumped CTDs with a sampling rate of 0.5 Hz and an
instrumental precision of 0.01 for salinity, 0.002 °C for temperature and 2.4 dbar for pressure.
CTD profiles are collected during an ascent from the parking depth to the surface, which takes
approximately 3 hours at a nominal vertical speed of 0.1 m/s.

The BGC-Argo float collected data for almost 2 years, between 23 June 2018 and 15
March 2020 (date of recovery during the PERLE 3 cruise). It remained in the centre of the
Tyrrhenian Basin (between 39.1-39.7° N, 11.6-12.8° E) during this period. The resulting time
series of CTD profiles includes 16 profiles between 0 and 1000 m from June 23 to July 6, 2018,
and 118 profiles between 0 and 2000 m from July 13, 2018 to March 15, 2020, with a time
resolution of 7 days until October 2019, then 3 days thereafter.

**2.4 Staircases detection**

Step structures were defined based on temperature and salinity profiles between 500 and
2000 m depth, following the procedure described in Durante et al. (2019). Relative maxima in
the vertical gradient of salinity and potential temperature are used to identify interfaces that
form well-marked steps and delimit a well-mixed layer. Gradient thresholds of 10-4°C/m for
potential temperature and 5 x10-4 PSU/m for salinity were used.



227

## 3. Results

### 3.1 Hydrological and hydrodynamical features along the section

#### 3.2.1 Temperature, salinity, oxygen, and nitrate

The temperature-salinity diagram (Fig. 1) and the basin cross-section (Fig. 2) clearly identify the distribution of the water masses present in the study area. As in Falco et al. (2016), we used the isohaline 38.72 as the minimum salinity value to identify the shallowest and deepest levels of the LIW. The MAW, which extends in the upper 200 m of the water column, has a higher temperature and lower salinity to the east. To the west, the colder TIW can be distinguished at about 150 m depth. The warmer, saltier, and oxygen-depleted core of the LIW is found at depths between 300 and 600 m. The core of the colder and less salty TDW is visible beyond 1200 m depth.

In March, the nitrate distribution in the Tyrrhenian Sea shows a nutrient-poor surface layer and a nutrient-rich deep layer, typical of oligotrophic conditions. Nitrate concentrations are more stable than the physical parameters, with only small variations across the section. Near the surface, nitrate concentrations are low, about 1 µmol/kg, due to biological uptake. With increasing depth, nitrate concentrations increase. At intermediate depths (250–650 m) they range from 4 to 7 µM, indicating a transition zone with maximum vertical gradients. In deeper waters, concentrations reach 7 to 9 µM due to decomposing organic matter.


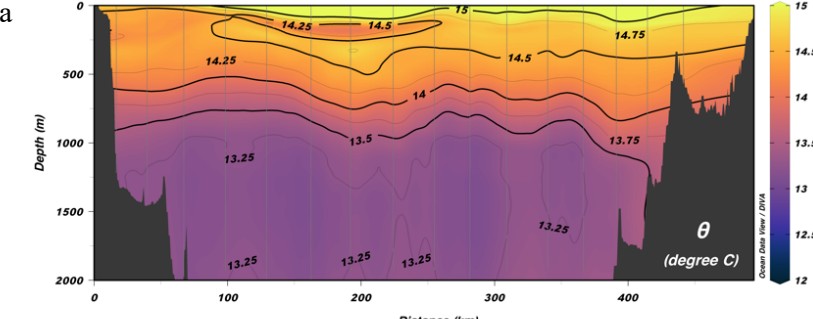



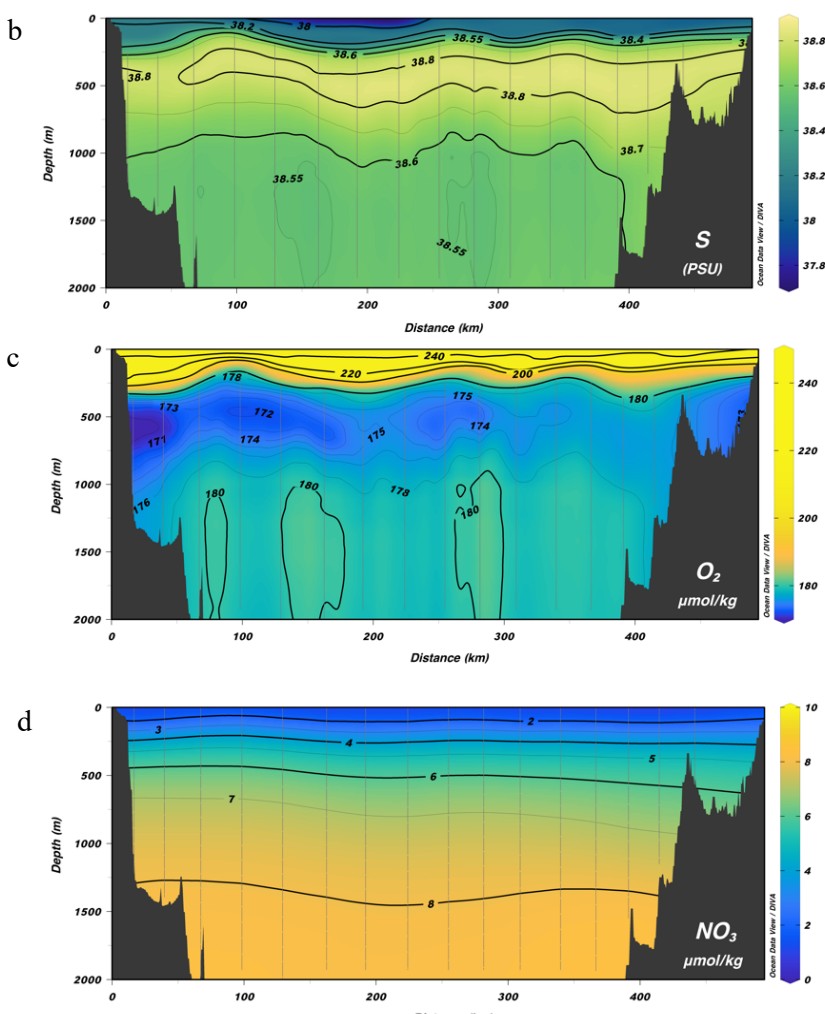

*Figure 2: Cross-basin section of (a) potential temperature, (b) salinity, (c) dissolved oxygen,*
*and (d) nitrates for the PERLE-3 cruise (March 2020).*

**3.1.2 Currents**
The dynamic topography and currents measured in the water column during PERLE 3
reveal the presence of two large mesoscale eddies in the centre of the basin with maximum
velocities of about ∼30 cm/s (Fig. 3). The cyclonic eddy at 10.5° E extends to a depth of about
200 m, while the eddy at 12° E extends to more than 500 m. The deep current running along
the eastern edge of the basin defines the general along-slope cyclonic circulation. Below the
LIW core, i.e. at 600 m depth, the currents are very weak, of the order of a few cm/s.





*Figure 3: (a) Mean absolute dynamic topography during the period of the cruise (14–16 March 2020) (b) Stick plot of vertically averaged S-ADCP currents between 21 m and 1200 m along the ship's route across the basin, (c) Meridional component of S-ADCP and L-ADCP currents between the 21 and 2000 m depth for the PERLE-3 cruise.*







Later in the text, we'll distinguish between deep stations without significant staircase
step structures, such as station 20, and stations with significant staircase steps, such as station 9.
In terms of vertical current velocities, the profiles show velocities on the order of mm/s, with
standard deviations of a few mm/s, as expected (Fig. 4).

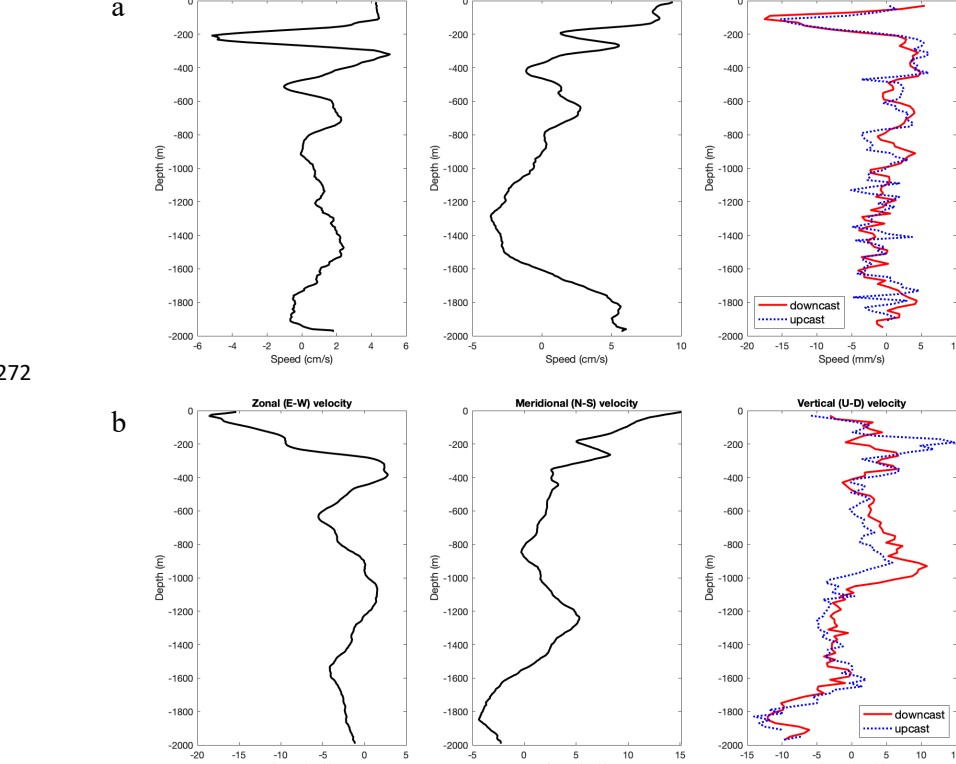



*Figure 4: Vertical profiles of the zonal and meridional components of the horizontal current,*
*and vertical velocities for (a) station 09 and (b) station 20 during the PERLE-3 cruise.*

**3.1.3 Turbidity, large particle abundance and Junge index**
The beam attenuation coefficient, an indicator of the abundance of small
particles (Fig. 5a) is highest in the surface layer and along the continental slope on both sides
of the basin. The tongue of turbid water that descends to 500–600 m in the western half of the
basin is associated with the downward movement of water around the anticyclonic eddy.




Coarse particles observed with the UVP (Fig. 5b) are most abundant along the
continental slope and between 400 and 900 m depth throughout the basin. The subduction effect
of the anticyclonic eddy is also evident in the abundance of large particles.

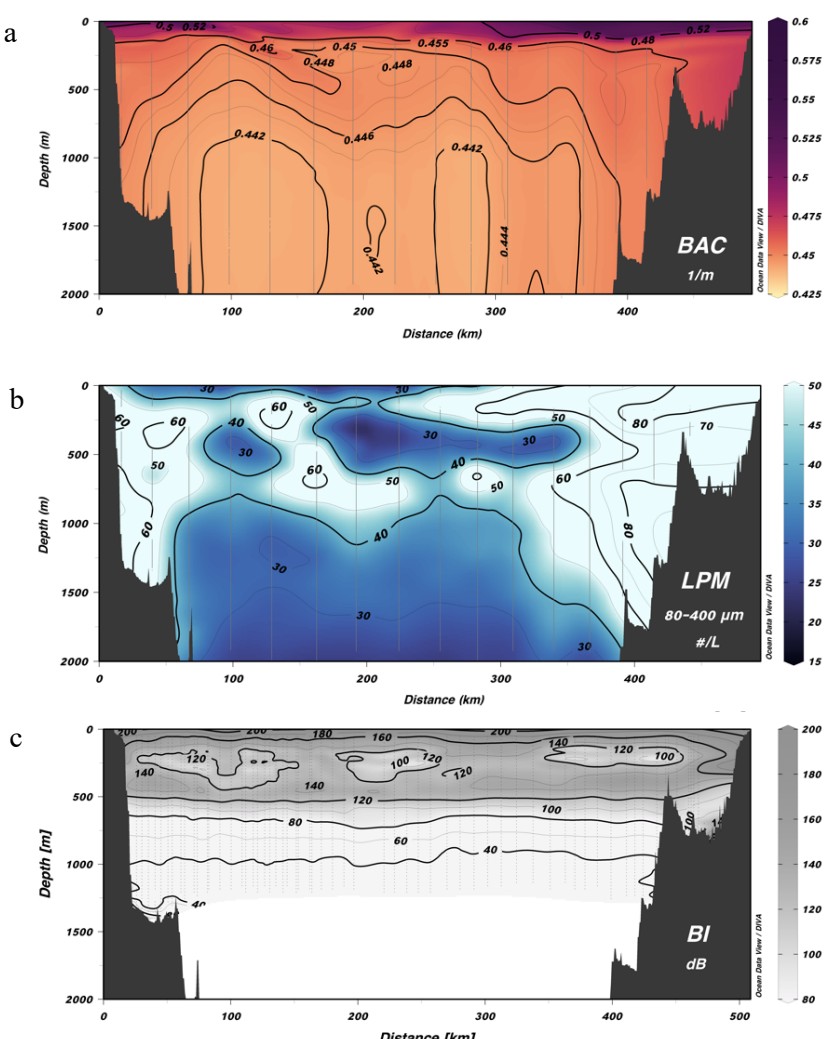



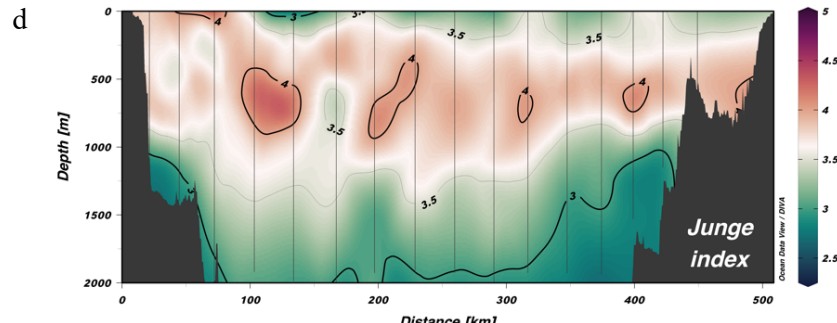

*Figure 5: Cross-basin section of (a) beam attenuation coefficient, (b) total large particle*
*abundances, (c) backscatter index for the PERLE-3 cruise, and (d) Junge index.*
An accumulation of large reflectors between 200 and 500 m depth is clearly visible on
the 38 kHz ADCP echo intensity (Fig. 5c). It corresponds to the deep scattering layer formed
by the micronekton. Its fragmentation is related to the diel vertical migrations of part of the
organisms.
The Junge index (Fig. 5d), estimated from UVP data, shows an intermediate maximum –
indicating the preponderance of smaller particles over coarser ones – between 400 and 1000 m,
below the deep scattering layer. Below 1000 m, the index decreases, indicating a relative
decrease in the abundance of smaller particles compared to coarser particles. This decrease is
more pronounced on the continental slope and below deep eddies.

## 298    3.2 Hydrological features of thermohaline staircases

### 299    3.2.1 Attributes of staircase station vs. non-staircase station

In this section we describe the vertical distribution of physical, dissolved and particulate
parameters at station 20, which shows virtually no significant staircase (Fig. 6), and station 09,
which shows marked staircases between 700 and 1700 m (Fig. 7). For all parameters, the
gradients observed at the level of the density steps are small, but significant given the resolution
of the sensors. For most of the variables at station 20, the profiles vary almost uniformly below
800 m depth, with the exception of a homogeneous layer that appears between 1070 and
1140 m, followed by a density step.
The profiles of physical variables at station 09 show a series of steps starting at 750 m
depth. Density steps result in thin interfaces (about 9–73 m) and density variations of a few
thousandths of a kg/m$^3$. The thickness of the mixed layers varies between 10 and 230 m and
increases significantly below 1000 m. The profiles of the biogeochemical variables also show
step-like profiles, with larger gradients corresponding to the density steps and nearly
homogeneous concentrations in the mixed layers between each step. This is clearly visible for
dissolved elements (oxygen, nitrates) and small particle concentration (beam attenuation
coefficient). The effect of the density steps on the abundance of coarser material is less obvious





due to the variability of the measurements, but it still appears that the total abundance decreases
significantly below each step. It is noteworthy that the decrease in the Junge index between 700
and 1600 m is greater for station 9 with staircase steps, of the order of one unit, than for
station 20, which is more irregular and of the order of 0.2 units.

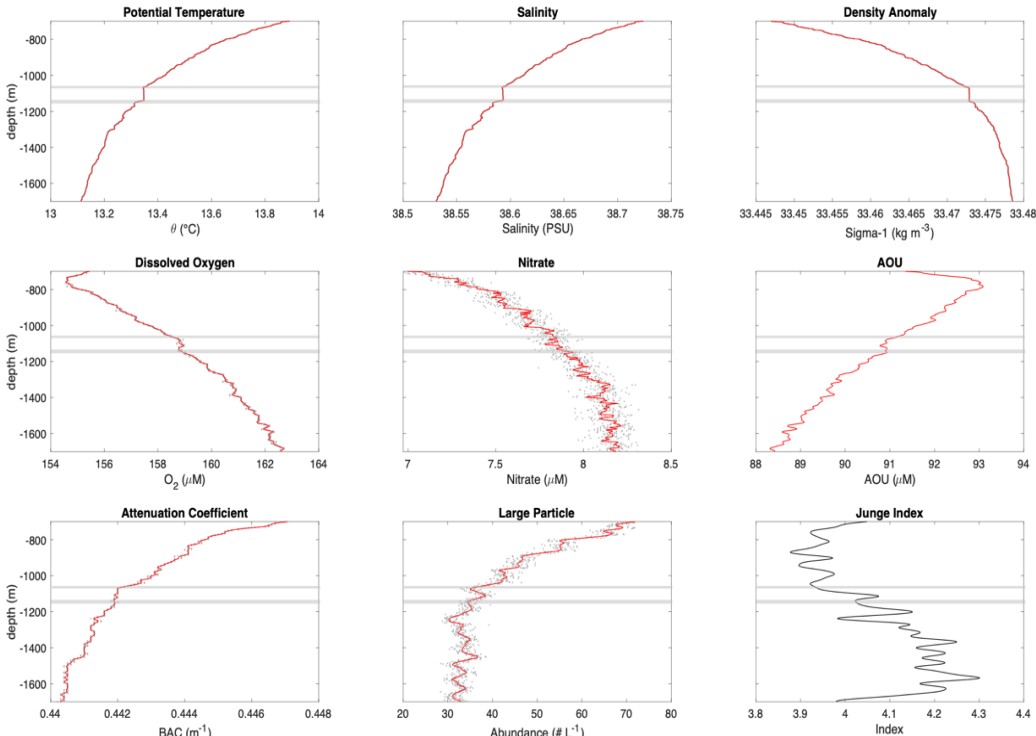

*Fig. 6: Profiles of potential temperature, salinity, potential density anomaly (top row) and*
*dissolved oxygen, nitrates, and apparent oxygen utilization (AOU) (middle row), beam*
*attenuation coefficient, large particle abundance between 80 and 400 µm, and Junge index*
*(bottom row) between 700 and 1700 m deep for station PERLE-3_20. The grey dots are the data*
*binned at 1-metre intervals and the solid red line indicates the smoothed profile.* See station
position in Fig. 1.



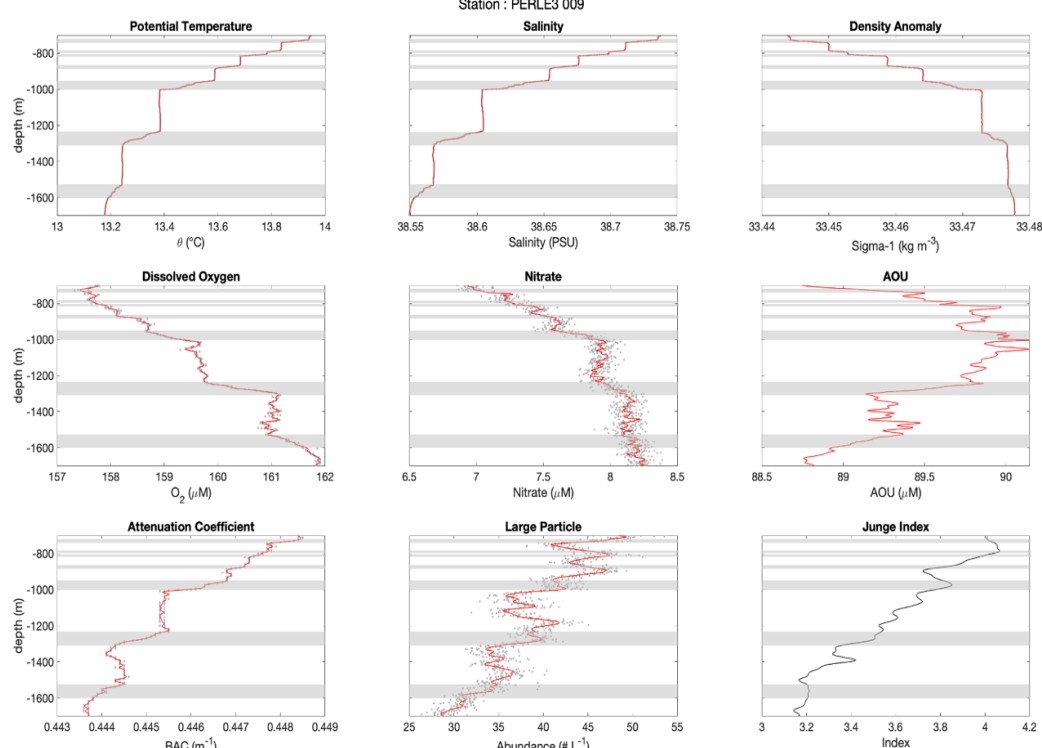

*Fig. 7: Profiles of potential temperature, salinity, potential density anomaly (top row) and*
*dissolved oxygen, nitrates, and apparent oxygen utilization (AOU) (middle row), beam*
*attenuation coefficient, large particle abundance between 80 and 400 μm, and Junge index*
*(bottom row) between 700 and 1600 m deep for station PERLE-3_09. The grey dots are the*
*data binned at 1-metre intervals and the solid red line indicates the smoothed profile. Major*
*density steps are delineated by horizontal grey lines.* See station positions in Fig. 1.

### 3.2.2 Positioning and persistence of main staircases

The transition zone between the LIW and the TDW thus provides the right conditions
(warmer and saltier water mass overlying a colder and less salty water mass) for the double
diffusion phenomenon by salt fingers. The analysis of the thermohaline gradients from the
profiles collected during the cruise and from the profiling float allowed us to identify and
position the main stepped structures in the Tyrrhenian Basin (Fig. 6).
During the PERLE3 cruise, the staircases develop clearly between 600 and 2000 m
depth, mostly in the centre of the basin (Fig. 8a). Staircases are absent under the deepest eddy



(about 12° E). Staircases are absent near the western slope and below the anticyclonic eddy at
about 12° E down to 1000 m depth.

The temporal evolution of the staircase in the centre of the basin, as seen by the profiling
float between July 2018 and March 2020, underscores that these structures, particularly at
depths greater than 1000 m, are relatively stable and have been maintained for several years
(Fig. 8b).

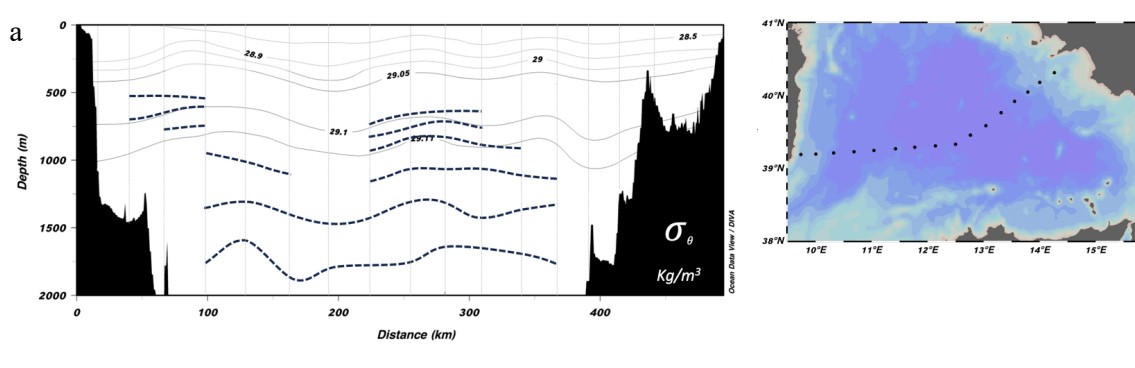

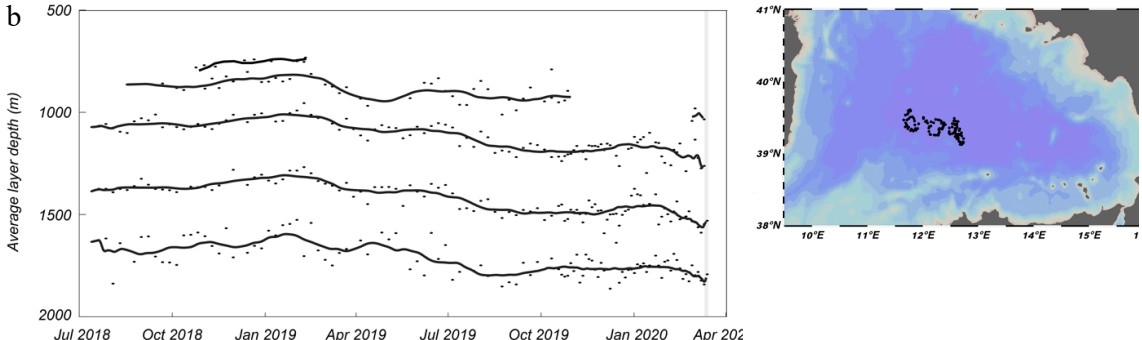


*Figure 8: (a) depth of the main thermohaline staircases across the basin during the PERLE3*
*cruise and (b) temporal evolution of the staircases in the centre of the basin as seen by the*
*profiling float between July 2018 and March 2020. The shaded band indicates the campaign*
*period. The position of the stations and the trajectory of the profiling float are shown on the*
*maps on the right.*

**4. Discussion**

**4.1 Characteristics and development conditions of the notable staircase**



A large part of the oceanic regions are suitable for the process of double diffusion and salt fingering, but it clearly develops only in some of them, especially in the Mediterranean. Meccia et al. (2016) have shown that about 50% of the Mediterranean Sea is favourable for double diffusion processes, with the Tyrrhenian, Ionian, and southwestern Mediterranean basins being most susceptible to salt fingering, and the strongest processes potentially occurring in the deep waters of the Tyrrhenian Sea.

Our observations in the Tyrrhenian Sea between the surface and 2000 m show some well-defined thermohaline staircase structures associated with salt fingers, while observations made during the PERLE cruises in the Ionian Sea and Levantine Basin do not show any particulate structures associated with salt fingers. The observed thermohaline staircases occur mainly in the centre of the Tyrrhenian Basin and are less defined on the continental slope at both ends of the section. These latter regions are the ones that contain the cyclonic boundary circulation that entrains the core of the LIWs originating from the Strait of Sicily. These results are consistent with those of Zodiatis and Gasparini (1996) from ship observations, of Buffett et al. (2017) from seismic observations, and Taillandier et al. (2020) from profiling float observations, who showed that the staircases with the most distinct step-like gradients appear in the centre of the basin, while they become more diffuse towards the boundaries and the bottom. They linked this change to increased vertical motions that prevent diffuse convection and staircase formation because the internal wave field and current shear are stronger near the boundaries.

Durante et al. (2021) suggested that mixing induced by internal gravitational waves can modulate the staircase structure. Our observations show that the presence of significant staircase structures down to 2000 m can also be influenced by mixing induced by cyclonic eddies present in the basin. When the vertical extent of these eddies reaches depths deeper than the LIW (i.e. about 500 m), the intensity and variability of the currents in the transition zone between the LIW and the TDW visibly alter the development of the staircases. These observations confirm that, even within the basin, step structures appear to develop preferentially in areas of weak horizontal and vertical currents. However, the modification by eddies would be episodic and would hardly affect the lower part of the salt finger region, as several studies show that the central Tyrrhenian thermohaline steps observed in the heart of the basin interior are quite stable and can persist for years (Taillandier et al., 2020) to decades (Durante et al., 2019).

### 4.2 Impact of staircases on the distribution and settling of particulate matter

The effect of density steps on the vertical distribution of particle abundance is detailed for three interfaces between 790 m and 950 m (Fig. 9) and one larger interface between 1150 and 1400 m (Fig. 10). Both examples show a significant reduction in the abundance of fine particles as seen by transmissometry (BAC) under each interface. For large particles seen by the UVP, the evolution of abundance on either side of the interface differs depending on size. For the smallest size ranges (80.6-128 µm) the abundance decreases slightly whereas for larger size ranges it hardly varies and, in some cases, even increases. This evolution with depth indicates that at each density interface, there is likely to be a retention of some of the finer particles, as well as coarser ones whose effective density is close to that of ambient water. The



larger particles, which are generally denser and have higher settling velocities, are not affected
by the change in water density.

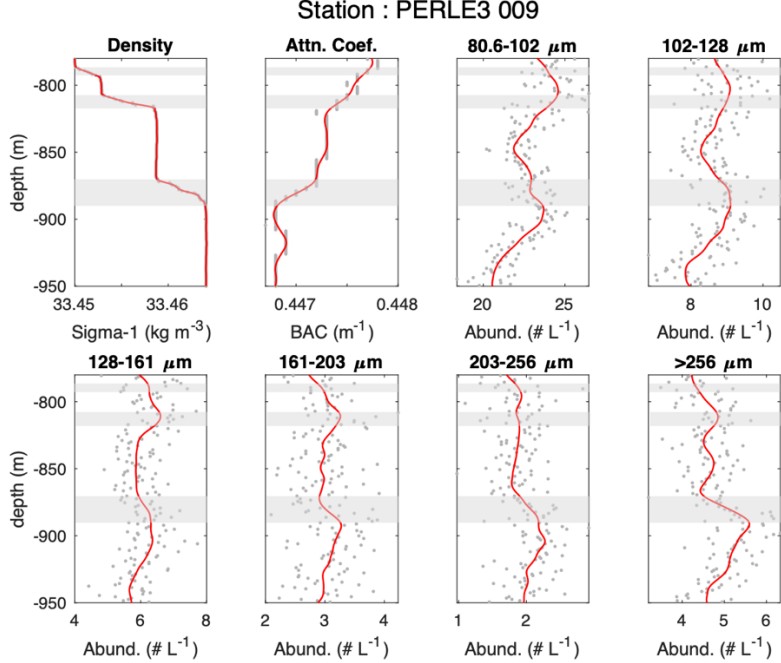


*Figure 9: Profiles of potential density anomaly, beam attenuation coefficient, large particle*
*abundances for the first four size classes and for all sizes>256 $\mu$m of the UVP between 780*
*and 950 m deep for station PERLE3-09. Grey stripes identify the main steps in density.*

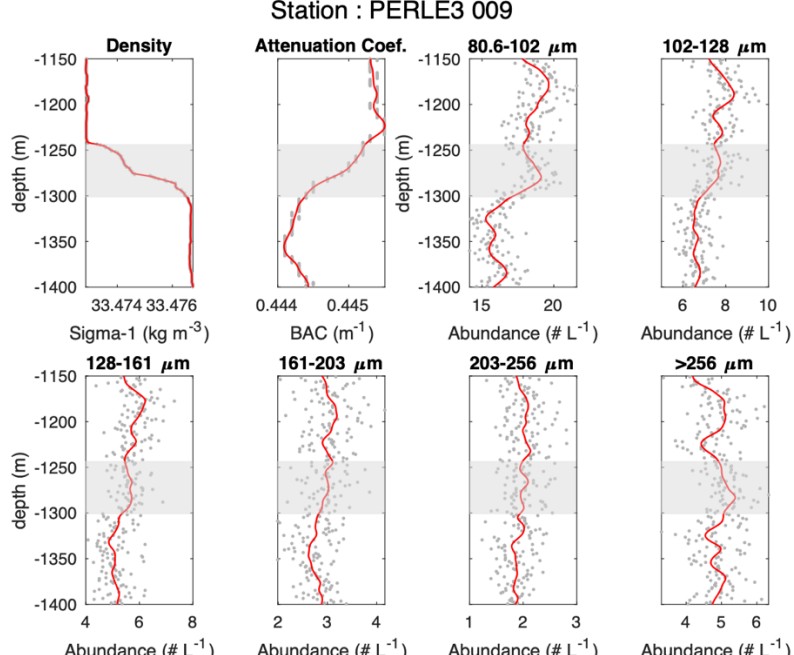


*Figure 10: Profiles of potential density anomaly, beam attenuation coefficient, large particle*
*abundances for the first four size classes and for all sizes>256 µm of the UVP between 1150*
*and 1400 m deep for station PERLE3-09. Grey stripes identify the main steps in density.*


The slope of the Junge-type particle size spectrum has been shown to be a valid first-
order description of the particle size distribution, particularly for the study of size-dependent
processes such as particle sinking (Guidi et al., 2009). In our case, the observed abundance of
coarse particles and the slope of the particle size spectrum are greatest in the deep scattering
layer between 400 and 800 m (Fig. 5). Deeper down to 2000 m, both the total particle
abundance and the slope of the size spectrum decrease, indicating that the contribution of
coarser particles to the particle population increases with depth. Similar results regarding the
size distribution of smaller particles in the Tyrrhenian Basin were obtained by Chaikalis et al.
(2021) during the trans-Mediterranean cruise in March 2018 using an in-situ laser scattering
and transmissometry instrument (LISST-Deep). In this study, they considered the size range
between 5.6 and 92.6 m, complementary to that measured with the UVP. Their stations show
similar profiles of the particle size spectrum below the surface layer (>150 m), with an
intermediate maximum at about 400–500 m depth and a steady decrease to 2000 m depth. Such
an evolution of the abundance and particle size spectra is generally considered to be the result
of aggregation, with smaller particles agglomerating to form larger ones, a common process in
the ocean (McCave, 1984).
Based on laboratory and modelling experiments in the literature, strong density
interfaces are known to cause particle retention and promote aggregation. According to





Doostmohammadi and Ardekani (2015), density interfaces can induce preferential retention of
fine, slow-sinking particles compared to larger particles, and can therefore promote the
formation of clusters of fine particles, which can then aggregate. This retention phenomenon
can also act on coarse particles, which are often formed by high porosity aggregates. According
to Kindler et al. (2010), large porous particles are much more dependent on transition zones
and large accumulations of porous particles can occur at interfaces due to their less dense water
transport, which is also conducive to aggregation. In our case, the staircase density interfaces
are really weak (a few thousandths of kg/m3) and despite the measurement uncertainty of the
particle abundances, the results suggest that these interfaces have an effect on the size
distribution – through aggregation – of the settling particles.

In addition, convection phenomena in the mixed layers sandwiching the density
interfaces are likely to even out the concentration of slowly sinking particles (especially the
finest and most numerous). The vertical velocities of the current measured, at ±a few mm/s, are
likely to alter the settling of particles, helping to equalise their abundance and increase their
residence time in each layer.

**4.3 Potential impact of staircases on the biogeochemical activity**

At station 20 (fig. 6), the lack of staircase structures and density interfaces results in
relatively uniform sedimentation of particulate matter and relatively homogeneous biogeo-
chemical processes throughout the water column. As particles descend into the deep ocean, they
undergo a progressive but overall constant mineralization process by bacteria (Ghiglione et al.,
2009). In contrast, stations with staircase structures show distinct layered profiles for dissolved
oxygen and nitrate (figs. 10 and 11). These staircases facilitate the retention and aggregation of
sinking particulate matter, which is critical for the mineralization and transformation of organic
matter in the water column (Wakeham and Lee, 1993). This remineralization process releases
additional nutrients and consumes oxygen, potentially leading to localized oxygen depletion or
an increase in Apparent Oxygen Utilization (AOU), particularly evident in the shallow stair-
cases between 750 and 900 m at station 9 (Fig. 11). At the deeper interface around 1300 m
(Fig. 12), the increase in AOU may be masked by a stronger vertical oxygen gradient. These
interfaces are thought to create specific microenvironments that influence microbial community
composition and local biological productivity, similar to the concept of plastispheres described
by Conan et al. (2022). Consequently, the formation of distinct layers allows various organisms
to colonize specific microhabitats, thereby enhancing biodiversity and trophic interactions.

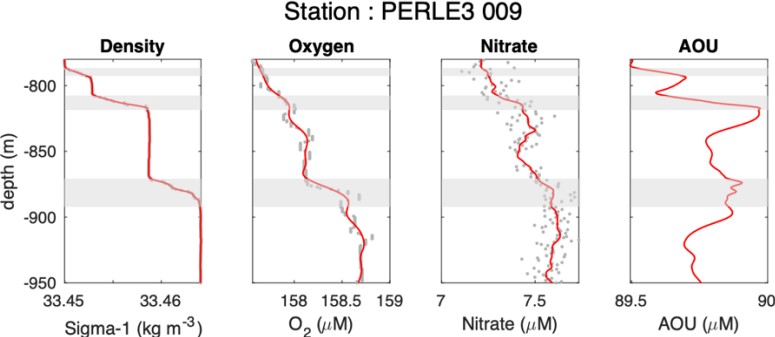


*Figure 11: Profiles of potential density anomaly, dissolved oxygen, nitrates and Apparent Oxygen Utilization (AOU) between 780 and 950 m deep for station PERLE3-09. Grey stripes identify the main steps in density.*

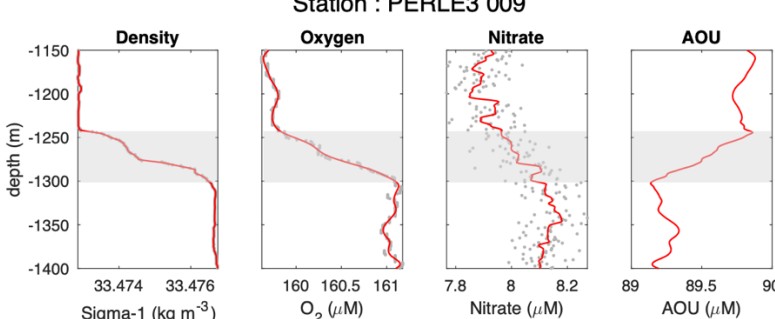

*Figure 12: Profiles of potential density anomaly, dissolved oxygen, nitrates and Apparent Oxygen Utilization (AOU) between 1150 and 1400 m deep for station PERLE3-09. Grey stripes identify the main steps in density.*

Taillandier et al. (2020) showed that the thermohaline staircases in the Tyrrhenian Sea
significantly influence the biogeochemical dynamics by contributing to the nitrate enrichment
of the LIW through diffusion. They estimated upward nitrate fluxes in the transition zone be-
tween 400 and 2000 m depth to be about 4 μmol/m²/d using the salt diffusivity formulation of
Radko and Smith (2012) based on the density ratio ($R\rho = -\alpha \cdot \partial\theta/\partial z - / -\beta \cdot \partial S/\partial z$), where $\alpha$ and
$\beta$ are the thermal expansion and haline contraction coefficients of seawater, respectively, and
the molecular diffusivity of heat is $k_T = 1.4 \times 10^{-7}$ m²/s. Based on this study, we calculated
nitrate fluxes at each interface using the same formulation. The nitrate flux can be expressed
as:

$F_{NO3} = K_{sf} \cdot \partial C_{NO3}/\partial z = k_T \cdot R_\rho \cdot (135.7/(R\rho-1)^{1/2} - 62.75) \cdot \partial C_{NO3}/\partial z$



At station 9, the diffusive nitrate flux was assessed for each depth step between 600 and
1600 m (Fig. 13). Fluxes ranged from 3.8 µmol/m²/d at the deepest depth (1254 m) to
19.1 µmol/m²/d for the three shallower depths (728 to 870 m). The fluxes at the interfaces are
therefore stronger that the overall flux estimated by Taillandier et al. (2020) for the 400-2000
m depth range. Fluxes at interfaces are the major contributor to the total nitrate flux. It is con-
ceivable that the release of additional nutrients at the upper interfaces increases the local vertical
gradient, thereby enhancing diffusive fluxes.

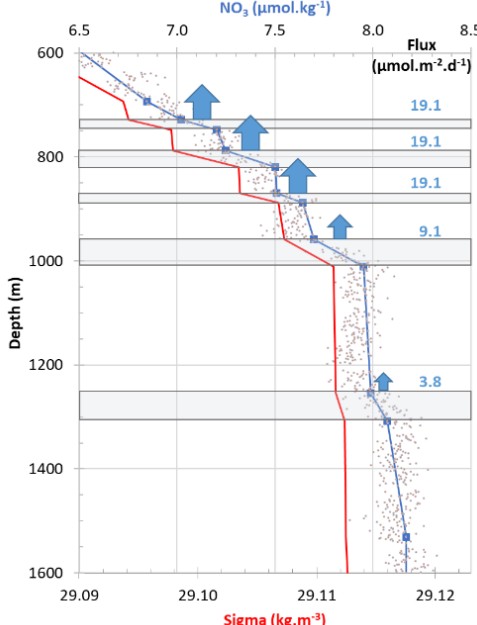


*Figure 13: Estimated diffusive fluxes of nitrate (in µmol/m²/d) from station 9 at the interfaces*
*of each gradient zone characterizing a staircase step. In red: the vertical profile of the specific*
*volume anomaly ($\sigma_\theta$ in kg/m³) delimiting homogeneous and gradient zones. The grey points*
*correspond to nitrate measurements by the SUNA, while the blue squares are the values*
*averaged over 1 m for each point at the upper and lower interfaces, making it possible to*
*calculate the nitrate gradient in the shaded areas.*

**5. Conclusion**

This study examines observations of the effect of weakly stepped density gradients
induced by salt fingering phenomena on the distribution and fluxes of particulate and dissolved
elements. While the effect of density steps on particle settling has been well studied
experimentally and demonstrated for strong density gradients in natural environments (i.e.
buoyant river plumes and pycnocline), this is the first time, to our knowledge, that this effect



has been identified for steps with the extremely low density gradients that exist in the deep ocean..

In the Tyrrhenian Sea, the interface between the warm and saline intermediate waters (LIW) and the colder and less saline deep waters (TDW) presents thermohaline conditions favourable to the development of significant salt fingering staircases. Profiles with mixed layers tens to hundreds of metres thick and density steps of a few thousandths of kg/m$^3$ occur in zones of low hydrodynamic energy, away from strong horizontal currents, such as near the boundaries. The study also shows that these steps are mostly stable and persist for years, especially in the central basin, while they are disrupted or absent in regions with stronger hydrodynamics, such as near deep-reaching eddies.

The presence of thermohaline staircases appears to significantly influence the distribution of particulate and dissolved matter. Profiles of particle abundance down to a few hundred microns, or profiles of dissolved substances such as oxygen and nitrates, tend to follow that of density, i.e. show staircases in areas affected by salt finger mixing processes, or vary regularly in unaffected areas.

The density steps caused by these staircases modify the particle size distribution, leading to an evolution towards larger aggregates. In fact, the retention of near-floating particles at density interfaces increases their residence time, promotes particle aggregation and, incidentally, allows the larger particles thus formed to cross the density interface, with possible implications for the vertical flux of carbon.

The staircases influence the concentration of nutrients (such as nitrates) and dissolved oxygen levels, thereby influencing the biogeochemical cycling within these layers. The retention of part of the particulate material at the density interfaces allows the mineralization of organic matter, the effect of which on oxygen and nitrate concentrations is superimposed on the upward flux by diapycnal diffusion. Ecologically, it is hypothesized that these staircases may play a crucial role in promoting diverse habitats and influencing the lability of organic matter and nutrient distribution.

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

**Data Availability Statement**
The   CTD   and   ADCP   data   used   in   this   study   are   available   at
https://doi.org/10.17600/18001342. The UVP data are available at https://ecopart.obs-vlfr.fr.

**Author Contributions**
Conceptualization and Methodology, XDDM, PB and MPP; Writing – Original Draft
Preparation, XDDM and PB; Writing – Review & Editing, All.

**Acknowledgements**
This study benefited from the data obtained during the PERLE-3 cruise with the R/V
Pourquoi-Pas? and was supported by the MISTRALS-MERMEX project.

**Competing interests**
The authors declare that they have no conflict of interest.