# Peer review of "Effect of double diffusion processes in the deep ocean on the distribution and dynamics of particulate and dissolved matter: a case study in Tyrrhenian Sea"

_EGUsphere, 2024_

## Referee Comment (RC1)

Review of EGUsphere-2024-3436

Effect of double diffusion processes in the deep ocean on the
distribution and dynamics of particulate and dissolved matter: a
case study in Tyrrhenian Sea
by Durrieu de Madron et al.

**Overview**

The manuscript by Durrieu de Madron et al. presents CTD, ADCP and optical profile data along a section crossing the Tyrrhenian Sea and investigates salt finger staircase effects on particulate and dissolved matter distribution.

I believe that the measurements would be interesting to the community. The analysis and results are generally well presented and the figures are clear. However, in some instances, the conclusions are fairly speculative, there are inconsistencies or redundancies, clarifications are lacking or phrasing is deficient. Therefore, I recommend a detailed revision based on the comments below.

**1 General comments**

a) I would suggest replacing 'salty' with 'saline' throughout

b) Replace 'shipborne' and 'ship-borne' with 'shipboard' throughout

c) Revise tenses for consistency; e.g., 'The first ADCP was' (line 180) [...] The second ADCP is' (line 182)

d) Add longitude as an upper x-axis on depth-distance section plots, for better reference to the map.

e) You could support the claims of staircase persistence with low measured microstructure derived dissipation rates and diapycnal diffusivity in Ferron et al. 2017 https://doi.org/10.1002/2017GL074169

**2 Specific comments**

Line 15: Replace 'will be' with 'is' to have a consistent tense

Line 45: ocean, temperature

Line 48: are on the order

Line 57: Replace 'Such thermohaline diffusive convection' with 'This' - diffusive convection refers to the double diffusive process opposite to salt finger formation.

Line 63: Perhaps add some references to support these numbers

Line 80: during their

Line 102: (Falco

Line 104: references

Line 108: cuts across

Line 119: water column with and without staircases, respectively, between

Line 122: , with depth in color.

Line 125: Shipboard

Line 129: with a shipboard rosette

Line 131: Profiler UVP

Line 133: 2 cm vertically

Line 134: °C and $4 \times 10^{-5}$? S/m, and

Line 135: °C and $3 \times 10^{-4}$? S/m, respectively.

Line 138: ] and resolution

Line 147: Define the quantities in the parenthesis. Maybe provide some details on the validation, e.g., how many samples over how many profiles.

Line 155: clarify what '$\#l^{-1}$' means

Line 159: and $-\alpha$ denotes ?

Line 163: represent up to

Line 176: scans

Line 186: 2 km given mean ship speed of ...

Line 187: various quality criteria (references?)

Line 187: Bathymetry (Etopo 1 resolution?) was incorporated in the processing to account for

Line 191: S-ADCP not defined (line 180?)

Line 199: downward looking

Line 203: which is a

Line 207: What are $w$ and $g$, how are all these parameters estimated? Perhaps give a bit more detail if you are going to include the equation.

Line 211: (I assume) CTD (singular)

Line 225 and 226: -4 in exponents

Line 261: Give source, date (or is it an average over the whole period?) and spatial resolution for mean absolute dynamic topography

Line 263: Clarify how this is both SADCP and LADCP?

Line 269: Delete 'In terms of vertical current velocities'

Line 326: What are the horizontal gray lines?

Line 338: Delete 'the double diffusion phenomenon by'

Line 339: Delete 'The analysis of the'

Line 340: Delete 'and position'

Lines 362-363: delete

Line 366: Replace 'strongest processes' with something more specific like 'thickest salt-finger-induced thermohaline staircase structures'

Line 371: Replace 'particulate' with 'particular'

Line 378: Replace 'more diffuse' with something like 'less evident'

Line 379: Replace 'diffuse convection' with 'diffusive convection'

Line 382: internal gravity waves

Lines 383-385: 'Our observations show that the presence of significant staircase structures down to 2000 m can also be influenced by mixing induced by cyclonic eddies present in the basin.' Please rephrase. This is not true as written since you did not measure or estimate mixing, it is just speculative based on the existence of steps / eddies.

Line 386: What is the evidence for 'the intensity and variability of the currents in the transition zone'?

Line 390: Unclear what is meant by 'the lower part of the salt finger region'

Line 391: Unclear what is meant by 'the central Tyrrhenian thermohaline steps observed in the heart of the basin interior' and how this relates to the PERLE observations.

Lines 417-432: Fig. 5 does not show a particle size spectrum, what is the evidence for these claims? Replace 'spectrum' with something like 'vertical distribution'.

Line 427: 92.6 $\mu$m ?

Line 428: similar particle distribution with depth below the ...

Line 442: m$^2$

Line 447: The current vertical velocity estimates on the order of mm/s [this is probably below what can be resolved]

Line 448: homogenize their abundance with depth

Line 484: delete 'Based on this study'

Line 487: define all the variables in the equation

Lines 478-501 and Fig. 13: Clarify that these are exclusively double diffusive fluxes, and they assume very small turbulence intensity (i.e., small turbulent diapycnal diffusivity).

Line 493-494: Not clear what is meant by 'It is conceivable that the release of additional nutrients at the upper interfaces increases the local vertical gradient, thereby enhancing diffusive fluxes.'

Line 497: each density vertical gradient

Line 500: Replace 'making it possible to' with 'and used to'

Line 501: add something like: 'Double diffusive nitrate fluxes across each interface are annotated in blue'

Line 504: examines the effect of observed weakly

Line 505: delete 'phenomena'

Line 520: resolved down to

Lines 522-523: i.e. are homogeneous at depths corresponding to thermohaline staircases.

Line 524: caused by salt finger staircases

Lines 532-533: matter, contributing to upward diapycnal diffusive fluxes of oxygen and nitrate.

---

## Referee Comment (RC2)

Reviewer #2

*Review of EGUsphere-2024-3436 paper: Effect of double diffusion processes in the deep ocean on the distribution and dynamics of particulate and dissolved matter: a case study in Tyrrhenian Sea by Xavier Durrieu De Madron and coauthors*

**General Comments**

This study presents and discusses various marine data collected in the spring of 2020 during the PERLE-3 cruise, and sheds light on the relationship between double diffusion, in the form of salt fingers, and particulate and dissolved matter in the Tyrrhenian Sea. The authors present and analyze hydrological, hydrodynamic, particulate, and dissolved data covering a section of the south-central Tyrrhenian Sea. They then focus on two different stations, one with and one without staircases due to salt fingers. They find that the steps influence the size and distribution of particulate and dissolved matter, which in turn can affect biological activities.

This study makes an interesting contribution to the growing understanding of the role of double diffusion processes in the ocean. The data are well characterized, the methods are clearly presented and the conclusions are innovative and substantial. However, some improvements could enhance both the text and the figures to increase the overall quality of the presentation. Moreover, it would be beneficial to publish the data.

**Specific Comments**

Introduction
- The organization of the introduction is somewhat challenging to follow, as it shifts from discussing particulate matter to staircases, then to sedimentation and to staircases again.
- The fourth paragraph (starting from line 64) seems unnecessarily lengthy, as it discusses plumes while the focus is on the deep ocean.
- It would be beneficial to include a brief overview of the circulation of water masses in the area, as this would help explain the local salt finger processes. Additionally, all acronyms used later in the text should be defined in the introduction.
- The third paragraph (starting from line 50) on double diffusion presents only one reference. Please add more references, particularly in the definition of the process (e.g., Radko has published numerous papers on the theoretical aspects of double diffusion).

Material and Methods

- Section 2.1: The title would sound better as "Thermohaline and Optical Data" (consider adding "and Derived Index"). It should also include both shipborne CTD data and Argo float data, as they represent the same type of data collected using different probes.

- Lines 149-150: The majority of parameters described seem to be unused in the subsequent analysis. Please highlight where and if they are utilized, or consider removing them.

- Transmissometer Paragraph (lines 152-168): This paragraph could be better structured. Additionally, what about particles in the 10 to 80 µm range?

- Section 2.2. (from line 178): I suggest titling this paragraph simply "Acoustical Data" to match the style of the preceding paragraph. Moreover, please define both L-ADCP and S-ADCP at the beginning and use consistent terminology for each type of ADCP.

Results Section 3.1

- Please ensure consistent use of physical units throughout the manuscript. For instance, both µmol and µM are employed in the initial paragraphs. I have attempted to highlight every discrepancy in the text revision for your reference.

- As previously mentioned, it would be beneficial to include a brief discussion in the introduction regarding the circulation and names of water masses in the Tyrrhenian Sea. Defining these terms solely in the captions detracts from the overall fluency of the text.

- Regarding the figures, it would enhance their legibility to incorporate references in the cross-basin section figures. For example, I suggest utilizing a different color to highlight the isolines mentioned in the text or integrating the names of the water masses, similar to the TS diagram (Fig. 1). Ideally, figures in scientific papers should be self-explanatory, presenting all necessary information for the reader to comprehend them.

- Please provide specific references for the subfigures. For example: "The warmer, saltier, and oxygen-depleted core of the LIW is located at depths between 300 and 600 m, as illustrated in Fig. 2 (sections a, b, and c)," each time you describe the illustrated parameters.

- In Figure 3, the units are missing. Additionally, in Figure 2, incorporating colored lines, symbols, or labels within the section could facilitate quicker identification of eddies.

- At line 267, stations 09 and 20 are introduced for the first time. It may be beneficial to present these stations earlier in the document (perhaps in section 2.1) and

provide a brief explanation for their selection (presumably because they are the most representative). Furthermore, consider adding the station names within Figure 4 to enhance reading clarity.

- In Figure 5, acronyms must be defined in the text upon their initial use before appearing in the figures (for instance, LPM has not been defined). As with the other results figures, please consider adding visual references to the two selected stations, such as corresponding lines on the cross sections.

- Lines 289-290: Is this statement conjecture or a well-established fact? If it is the latter, please provide a citation to support the claim. Additionally, it would be prudent to include a brief definition of micronekton, as it has not been mentioned previously, and the intended scientific audience may include researchers in abiotic studies.

Results Section 3.2

- Lines 316-318: Could you please provide additional explanation for this highlight?

- Figure 6: I recommend enlarging the text size, as it may be challenging to read depending on the medium used by readers.

- Line 341: Did you intend to reference Figure 8 instead?

- Lines 343-346: Please rephrase this sentence for clarity, as it currently lacks precision.

- Lines 284-351: The temporal and spatial stability of Tyrrhenian staircases has been documented in several studies (such as Johannessen and Lee, 1974; Molcard and Williams, 1975; Molcard and Tait, 1977; Zodiatis and Gasparini, 1996; Falco et al., 2016; Durante et al., 2019; and Taillandier et al., 2020; and more). It would be advantageous to highlight this consistency with existing literature, or alternatively, to include this information in the Introduction section.

- Additionally, please consider incorporating the graphical positions of the two eddies in Figure 8 and adding titles to the maps, such as "Cruise" and "Floats."

Discussion 4.1

The content in paragraphs 382-393 contains qualitative speculation and should be rewritten, as it lacks sufficient adherence to the presented results.

It is important to distinguish between what constitutes a measured result (and where to locate it in the previous text) and what represents a conjecture. While I understand this paragraph relates to Results Section 3.2.2, both sections require restructuring and more substantive argumentation to enhance coherence. Consider these questions: Can you demonstrate a definitive influence of the eddies on staircase structures, or is it merely a qualitative correlation? Did you select station 20 due to its proximity to the 12 E° eddy, with the intention of investigating the relationship between the eddy and the potential

development of steps? If this is the case, please state it explicitly to clarify your speculative assertions regarding their interaction.

Additionally, you mention that vertical currents behave as expected (line 270). In what manner can they visibly alter the formation of staircases (line 387), particularly since there are no staircases present at the station nearest the eddy? Can you compare your cast in station 20 with other previous data of the same station? A potential approach could involve starting from the persistence of the center-basin staircases and hypothesizing that the absence of staircases at your station 20 (and possibly adjacent stations, are you able to show them for comparison?) may also be attributable to the presence of the eddies and their associated stronger currents. This interpretation reflects my understanding of this segment of the discussion.

For your reference, a relevant study examining the relationship between a Meddy and staircases is presented in Hebert (1988).

**Discussion 4.2**
Please consider including a description of the size classes presented in either the text or a table for clarity.

**Discussion 4.3**
Lines 488-489: Since you are comparing your results with those of Taillandier et al. (2020), it would be beneficial to also reference Durante et al. (2021) in this paragraph. The study analyzed heat and salt fluxes of staircases in a nearby area and over a broader portion of the water column, yet it found the strongest thermohaline fluxes occurring within a similar depth range (700 m – 1600 m) as the one you selected.

**Technical Comments**
Please refer to the attached paper. Consider these comments as a complement to Reviewer #1's feedback.

---

## Referee Comment (RC3)

[referee-annotated manuscript omitted]

---

## Author Response (AR1)

**Author's response**

We would like to thank the three reviewers for their constructive criticism and comments, which enabled us to correct and improve the manuscript.

We have taken into account all suggestions for improving the text.

We have modified and improved the figures according to the reviewers' suggestions. Some figures (vertical velocities, distribution of physical and biogeochemical parameters at the level of individual interfaces for several stations) are new. A table on the abundance of the different particle size classes observed by the UVP has been added as supplementary data.

We have modified the introduction by deleting a paragraph on double sediment diffusion, which is active in areas with very strong gradients, and by completing another paragraph by providing the hydrological and hydrodynamic context of the region.

We have restructured the Materials and methods section by grouping together, as suggested, all the hydrological and optical measurements made during the PERLE3 campaign and by the profiling float. A new paragraph on the selection of representative stations has been added at the end of this section.

The results section has retained the same structure but has been expanded in response to the reviewers. Instead of showing vertical velocity profiles for two stations, we have chosen to illustrate the variability of vertical velocities for all stations with a histogram.

The results section retains the same three subsections, dealing respectively with circulation and thermohaline staircases, the effect of staircases on particulate matter distribution and deposition, and the potential effect of staircases on biogeochemical activity.

The first subsection has been reworded to link our observations to the literature, to modulate the effect of eddies on staircases, and to emphasise the persistence of thermohaline staircases at depths greater than 1000 m.

Three new figures detailing the evolution of physical, particulate and biogeochemical parameters at density interfaces for three different stations (7, 9 and 11) now replace the examples originally chosen for station 9. They are used to illustrate the other two subsections.

The second subsection on particulate matter has been reworded to emphasise the effect of density gradients on particulate matter dynamics.

The third part on biogeochemistry has been expanded to better explain the degradation processes that can occur at these interfaces and the consequences for nitrate fluxes.

Part of the conclusion has been reworded.

A number of references have been added.

**Responses to comments from Reviewer #1**

**Overview**

The manuscript by Durrieu de Madron et al. presents CTD, ADCP and optical profile data along a section crossing the Tyrrhenian Sea and investigates salt finger staircase effects on particulate and dissolved matter distribution. I believe that the measurements would be interesting to the community. The analysis and results are generally well presented and the figures are clear. However, in some instances, the conclusions are fairly speculative, there are inconsistencies or redundancies, clarifications are lacking or phrasing is deficient. Therefore, I recommend a detailed revision based on the comments below.

We thank the reviewer for his constructive comments. We have tried to respond to all of his comments and have provided detailed answers to his questions. A number of references have been added or changed in response to suggestions. Note that the names of the water masses described in the manuscript have been modified to be in line with the latest recommendations published in the article by Schroeder et al. (2024) A consensus-based, revised and comprehensive catalog for Mediterranean water masses acronyms. Mediterranean Marine Science, Vol. 25 No. 3, 783–791.

**1 General comments**

- a) I would suggest replacing 'salty' with 'saline' throughout Done
- b) Replace 'shipborne' and 'ship-borne' with 'shipboard' throughout Done
- c) Revise tenses for consistency; e.g., 'The first ADCP was' (line 180) [...] The second ADCP is' (line 182)

The concordance of tenses has been checked throughout the text.

d) Add longitude as an upper x-axis on depth-distance section plots, for better reference to the map.

The position of stations has been added at the top of each section, making it easier to link to the map.

e) You could support the claims of staircase persistence with low measured microstructure derived dissipation rates and diapycnal diffusivity in Ferron et al. 2017 https://doi.org/10.1002/2017GL074169

This reference was used in the first part of the discussion dealing with the circulation and thermohaline staircases.

**2 Specific comments**

Line 15: Replace 'will be' with 'is' to have a consistent tense Corrected

Line 45: ocean, temperature

Corrected

Line 48: are on the order

Corrected

Line 57: Replace 'Such thermohaline diffusive convection' with 'This' – diffusive convection refers to the double diffusive process opposite to salt finger formation.

Corrected

Line 63: Perhaps add some references to support these numbers

We cite the paper by van der Boog et al (2021) which provides temperature and salinity ranges at the interface levels of thermohaline staircases.

Line 80: during their

Corrected

Line 102: (Falco *Corrected*

Line 104: references

We added several references showing the presence of Tyrrhenian thermohaline staircases.

Line 108: cuts across

Corrected

Line 119: water column with and without staircases, respectively, between

Corrected

Line 122:, with depth in color.

Corrected

Line 125: Shipboard

**Corrected**

Line 129: with a shipboard rosette

Modified by 'mounted on a rosette carrying 22 twelve-liter Niskin bottles'

Line 131: Profiler UVP

Corrected

Line 133:2 cm vertically

Corrected

Line 134:  $^{\circ}$ C and 4 × 10 – 5? S/m, and Corrected:  $-2.10^{-4}$   $^{\circ}$ C (4.10-5 S/m), and

Line 135: °C and  $3 \times 10 - 4$ ? S/m, respectively. Corrected:  $2.10^{-3}$  °C  $(3.10^{-4}$  S/m), respectively.

Line 138:] and resolution

Corrected

Line 147: Define the quantities in the parenthesis. Maybe provide some details on the validation, e.g., how many samples over how many profiles.

Corrected. The sentence now reads 'A good correlation was observed with bottle measurements over the entire water column collected during the cruise at station PERLE3-10 ( $R^2$ = 0.99, N=27 samples,  $p < 10^{-5}$ , NO3SUNA = 0.991× NO3Bul – 0.116).'

Line 155: clarify what '#I-1' means

Corrected. # $L^{-1}$  means number of particles per litre.

Line 159: and  $-\alpha$  denotes?

Corrected. The missing parameter has been added.

Line 163: represent up to

Corrected.

Line 176: scans *Corrected.*

Line 186:2 km given mean ship speed of...

Short-term averaged data (2 min) were used to calculate the mean horizontal current every 2 km along the ship's path, regardless of the ship's speed.

Line 187: various quality criteria (references?)

Corrected. The sentence now reads 'various quality criteria (i.e., thresholds on vertical velocity error, vertical shear, correlation, minimum percentage of valid ensembles, Kermabon et al., 2018).'

Line 187: Bathymetry (Etopo 1 resolution?) was incorporated in the processing to account for Corrected. The sentence now reads 'Bathymetry (Etopo 1 with 1 arcminute resolution) was incorporated in the processing to account for bottom detection.'

Line 191: S-ADCP not defined (line 180?)

Corrected. This acronym is now defined as it first appeared in the previous paragraph.

Line 199: downward looking

Corrected.

Line 203: which is a

Corrected.

Line 207: What are w and g, how are all these parameters estimated? Perhaps give a bit more detail if you are going to include the equation.

The equation has been corrected. w and g are simply indices to differentiate between losses due to absorption in water (w) and geometric dispersion (g).

Line 211: (I assume) CTD (singular)

Right. Corrected.

Line 225 and 226:-4 in exponents

Corrected.

Line 261: Give source, date (or is it an average over the whole period?) and spatial resolution for mean absolute dynamic topography

A sentence has been added in the caption of Fig. 3: 'The product is the European Seas Gridded L 4 Sea Surface Heights and Derived Variables product, interpolated to a 3.75 arcmin grid, provided by the Copernicus Marine Environment Monitoring Service (CMEMS).'

Line 263: Clarify how this is both SADCP and LADCP?

Corrected. The legend clearly states that this is the combined S-ADCP and L-ADCP data.

Line 269: Delete 'In terms of vertical current velocities'

Done.

Line 326: What are the horizontal gray lines?

The horizontal grey lines highlight the interfaces of the thermohaline staircases. It has been clarified in the legend.

Line 338: Delete 'the double diffusion phenomenon by'

Done.

Line 339: Delete 'The analysis of the'

Done.

Line 340: Delete 'and position'

Done.

Lines 362-363: delete

Done.

Line 366: Replace 'strongest processes' with something more specific like 'thickest salt-finger-induced thermohaline staircase structures'

Corrected.

Line 371: Replace 'particulate' with 'particular'

Corrected.

Line 378: Replace 'more diffuse' with something like 'less evident'

Corrected.

Line 379: Replace 'diffuse convection' with 'diffusive convection'

Corrected.

Line 382: internal gravity waves

Corrected.

Lines 383–385: 'Our observations show that the presence of significant staircase structures down to 2000 m can also be influenced by mixing induced by cyclonic eddies present in the basin.'

Please rephrase. This is not true as written since you did not measure or estimate mixing, it is just speculative based on the existence of steps/eddies.

We agree that this statement is based solely on the co-occurrence of the presence of the eddy and the absence of thermohaline staircases, and that there is not necessarily a causal relationship. However, we could argue for this effect based on the results recently published by Yang et al. (2024), who showed that the presence of thermohaline staircases in the Caribbean Sea is often perturbed by subsurface mesoscale eddies. We have rephrased our discussion to this topic and only suggested possible interactions between thermohaline staircases and oceanic processes based on this work.

In this section we mostly focus on the spatial and temporal stability of thermohaline staircases in the central part of the basin (according to our observations and the literature); a stability that is likely to have a notable effect on the settling and transformation of particulate and dissolved material, which we detail in the following sections.

Yang, S., Zhang, K., Song, H. et al. Disruptions in thermohaline staircases caused by subsurface mesoscale eddies in the eastern Caribbean Sea. Nature Communications Earth & Environment 5, 408 (2024). https://doi.org/10.1038/s43247-024-01577-3

Line 386: What is the evidence for 'the intensity and variability of the currents in the transition zone'? We have no direct evidence, from ADCP measurements during the cruise, of an intensification of currents variability in the transition zone below the eddy. Our hypothesis was that the effects of a deep eddy could intensify turbulence and disrupt the development of thermohaline staircases. However, in the absence of additional data to support this hypothesis, we decided to modulate it, and only recall, based on the work of Yang et al (2024), the idea that eddy could interact with thermohaline staircases.

Line 390: Unclear what is meant by 'the lower part of the salt finger region'

By the lower part, we mean the region at a depth of more than 1000 m. This entire section has been rewritten.

Line 391: Unclear what is meant by 'the central Tyrrhenian thermohaline steps observed in the heart of the basin interior' and how this relates to the PERLE observations.

Previous studies (Zodiatis and Gasparini, 1996; Falco et al., 2016; Durante et al., 2019; Taillandier et al., 2020) confirm the observations obtained during the PERLE cruise in the sense that thermohaline staircases are strongest in the central part of the Tyrrhenian basin and are weaker or absent along the edges of the basin. We modified the text accordingly.

Lines 417–432: Fig. 5 does not show a particle size spectrum, what is the evidence for these claims? Replace 'spectrum' with something like 'vertical distribution'.

We are actually interested in the variation of the slope of the particle size distribution or spectrum, and do not present size spectra as such. The calculation and significance of the slope of the particle size spectrum is explained in section 2.1 on UVP data. We have corrected the terminology in the text.

Line 427:92.6 μm?

Right. Corrected.

Line 428: similar particle distribution with depth below the...

Corrected.

Line 442: m2

Rather than indicating vertical gradients, here we indicated the density difference at the interface between two staircases.

Line 447: The current vertical velocity estimates on the order of mm/s [this is probably below what can be resolved]

Figure 4 shows a histogram of the vertical velocities recorded over the whole section. The velocities vary between +15 and -15 mm/s with a rms of 4 mm/s. Such velocities are of the order of what can be solved with L-ADCP.

Line 448: homogenize their abundance with depth

Corrected.

Line 484: delete 'Based on this study'

Corrected.

Line 487: define all the variables in the equation

Done

Lines 478–501 and Fig. 13: Clarify that these are exclusively double diffusive fluxes, and they assume very small turbulence intensity (i.e., small turbulent diapycnal diffusivity). *Done*

Line 493–494: Not clear what is meant by 'It is conceivable that the release of additional nutrients at the upper interfaces increases the local vertical gradient, thereby enhancing diffusive fluxes.'

We have now clarified this idea by changing the previous sentence by the paragraph: Indeed, we hypothesize that the increase in microbial mineralization activity, occurring within the context of thermohaline staircases, releases nitrates among others. These nitrates can temporarily accumulate, creating bubble-like pockets of higher concentration (i.e. enriched microenvironments). The resulting increase in the nitrate concentration gradient enhances diffusive upward fluxes (as the underlying waters are richer).

Line 497: each density vertical gradient

Done

Line 500: Replace 'making it possible to' with 'and used to'

Done

Line 501: add something like: 'Double diffusive nitrate fluxes across each interface are annotated in blue' *According to the remarks, the legend is now:*

Estimated double diffusive nitrate fluxes across each interface (annotated in blue and expressed in  $\mu$ mol/m²/d) from station 9. The blue arrows indicate the direction of the flux and sizes are proportional to the intensity. In red: the vertical profile of the specific volume anomaly ( $\sigma\theta$  in kg/m-3) delimiting homogeneous and gradient zones. The grey points correspond to nitrate measurements by the SUNA, while the blue squares are the values averaged over 1 m for each point at the upper and lower interfaces, and used to calculate the nitrate gradient in the shaded areas.

Line 504: examines the effect of observed weakly

Corrected.

Line 505: delete 'phenomena'

Corrected.

Line 520: resolved down to

Corrected.

Lines 522–523: i.e. are homogeneous at depths corresponding to thermohaline staircases.

Corrected.

Line 524: caused by salt finger staircases

Corrected.

Lines 532–533: matter, contributing to upward diapycnal diffusive fluxes of oxygen and nitrate.

Corrected.

**Responses to comments from Reviewer #2**

Review of EGUsphere-2024-3436 paper: Effect of double diffusion processes in the deep ocean on the distribution and dynamics of particulate and dissolved matter: a case study in Tyrrhenian Sea by Xavier Durrieu De Madron and coauthors

**General Comments**

This study presents and discusses various marine data collected in the spring of 2020 during the PERLE-3 cruise, and sheds light on the relationship between double diffusion, in the form of salt fingers, and particulate and dissolved matter in the Tyrrhenian Sea. The authors present and analyze hydrological, hydrodynamic, particulate, and dissolved data covering a section of the south-central Tyrrhenian Sea. They then focus on two different stations, one with and one without staircases due to salt fingers. They find that the steps influence the size and distribution of particulate and dissolved matter, which in turn can affect biological activities.

This study makes an interesting contribution to the growing understanding of the role of double diffusion processes in the ocean. The data are well characterized, the methods are clearly presented and the conclusions are innovative and substantial. However, some improvements could enhance both the text and the figures to increase the overall quality of the presentation. Moreover, it would be beneficial to publish the data.

We thank the reviewer for his constructive comments. We have tried to respond to all his suggestions and we have detailed our answers to his questions.

**Specific Comments**

**Introduction**

• The organization of the introduction is somewhat challenging to follow, as it shifts from discussing particulate matter to staircases, then to sedimentation and to staircases again.

We have modified the introduction by deleting a paragraph on double sediment diffusion, which is active in areas with very strong gradients, and by completing another paragraph by providing the hydrological and hydrodynamic context of the region.

• The fourth paragraph (starting from line 64) seems unnecessarily lengthy, as it discusses plumes while the focus is on the deep ocean.

This paragraph has been removed

• It would be beneficial to include a brief overview of the circulation of water masses in the area, as this would help explain the local salt finger processes. Additionally, all acronyms used later in the text should be defined in the introduction.

An overview of water masses and general circulation in the Tyrrhenian Sea has been added. The names of the water masses described in the manuscript have been modified to conform to the latest recommendations published in the article by Schroeder et al. (2024) A consensus-based, revised and comprehensive catalogue for Mediterranean water masses acronyms. Mediterranean Marine Science, Vol. 25 No. 3, 783–791.

• The third paragraph (starting from line 50) on double diffusion presents only one reference. Please add more references, particularly in the definition of the process (e.g., Radko has published numerous papers on the theoretical aspects of double diffusion).

A couple of relevant references to articles and reviews on double diffusion have been added.

**Material and Methods**

• Section 2.1: The title would sound better as 'Thermohaline and Optical Data' (consider adding 'and Derived Index'). It should also include both shipborne CTD data and Argo float data, as they represent the same type of data collected using different probes.

We have changed the title of the paragraph and integrated the data collected by the profiling float as suggested. We have also included the paragraph on staircase detection.

• Lines 149–150: The majority of parameters described seem to be unused in the subsequent analysis. Please highlight where and if they are utilized, or consider removing them.

The parameter names have been corrected.

• Transmissometer Paragraph (lines 152–168): This paragraph could be better structured. Additionally, what about particles in the 10 to 80  $\mu$ m range?

The paragraph was split to distinguish the data from the transmissometer from that obtained from the UVP. We described in more detail the data collected by the transmissometer. The transmissometer measures all the particles that pass through the beam, but because of the abundance of the finest particles, it is more sensitive to them.

Section 2.2. (from line 178): I suggest titling this paragraph simply 'Acoustical Data' to match the style
of the preceding paragraph. Moreover, please define both L- ADCP and S-ADCP at the beginning and
use consistent terminology for each type of ADCP.

We have changed the title of the paragraph as suggested. The acronyms L- ADCP and S-ADCP have been clarified.

**Results Section 3.1**

• Please ensure consistent use of physical units throughout the manuscript. For instance, both μmol and μM are employed in the initial paragraphs. I have attempted to highlight every discrepancy in the text revision for your reference.

We have checked the consistency of the units throughout the text and the figures.

• As previously mentioned, it would be beneficial to include a brief discussion in the introduction regarding the circulation and names of water masses in the Tyrrhenian Sea. Defining these terms solely in the captions detracts from the overall fluency of the text.

A presentation of the main water masses and general circulation in the Tyrrhenian Sea has been added to the introduction.

Regarding the figures, it would enhance their legibility to incorporate references in the cross-basin
section figures. For example, I suggest utilizing a different color to highlight the isolines mentioned in
the text or integrating the names of the water masses, similar to the TS diagram (Fig. 1). Ideally, figures
in scientific papers should be self-explanatory, presenting all necessary information for the reader to
comprehend them.

The names of the water masses have been integrated into the temperature and salinity cross-section figures.

• Please provide specific references for the subfigures. For example: 'The warmer, saltier, and oxygen-depleted core of the LIW is located at depths between 300 and 600 m, as illustrated in Fig. 2 (sections a, b, and c),' each time you describe the illustrated parameters.

Thanks, we provided specific references for the subfigures throughout the text.

• In Figure 3, the units are missing. Additionally, in Figure 2, incorporating colored lines, symbols, or labels within the section could facilitate quicker identification of eddies.

Figure 3 has been corrected to add the missing units, a reference for geostrophic velocities. The legend now states the origin of the altimeter products.

• At line 267, stations 09 and 20 are introduced for the first time. It may be beneficial to present these stations earlier in the document (perhaps in section 2.1) and provide a brief explanation for their selection (presumably because they are the most representative). Furthermore, consider adding the station names within Figure 4 to enhance reading clarity.

Indeed, we chose these two stations as representative of the most contrasting profiles, i.e. with and without marked staircases. Subsequently, in the discussion section 4.2 Effect of staircases on particle distribution and settling, we present several examples based on zooms of staircase structures for three different stations (stations 8, 9 and 11).

We have provided a brief explanation of their selection in a new section at the end of the material and method section (2.3 Selection of representative stations).

• In Figure 5, acronyms must be defined in the text upon their initial use before appearing in the figures (for instance, LPM has not been defined). As with the other results figures, please consider adding visual references to the two selected stations, such as corresponding lines on the cross sections.

Corrected. The acronyms BAC and LPM were defined when they first appeared in the text.

• Lines 289–290: Is this statement conjecture or a well-established fact? If it is the latter, please provide a citation to support the claim. Additionally, it would be prudent to include a brief definition of micronekton, as it has not been mentioned previously, and the intended scientific audience may include researchers in abiotic studies.

The contribution of the micronekton to the deep scattering layer is well known. More detail and references have been added in the text.

**Results Section 3.2**

• Lines 316–318: Could you please provide additional explanation for this highlight?

This observation of a decrease in the Junge parameter as a function of depth, indicating a greater relative abundance of larger particles compared to finer ones, has been modified. While the decrease is significant and comparable for all stations between 800 and 2000 m depth, some stations, such as station 20, show smaller local variations between 1000 and 1600 m. The reasons for this remain unexplained, so we have removed this point from the text.

• Figure 6: I recommend enlarging the text size, as it may be challenging to read depending on the medium used by readers.

**Corrected**

• Line 341: Did you intend to reference Figure 8 instead?

**Indeed. Corrected**

• Lines 343–346: Please rephrase this sentence for clarity, as it currently lacks precision.

Corrected. The sentence now reads, 'During the PERLE-3 cruise, distinct thermohaline staircases are observed between 600 and 2000 m depth, primarily in the central region of the basin (Fig. 8a). These staircases are notably absent in two areas: near the western slope of the basin and beneath the deep anticyclonic eddy located at about 12° E. The absence of staircases extends to 1000 m depth beneath this eddy.'

• Lines 284–351: The temporal and spatial stability of Tyrrhenian staircases has been documented in several studies (such as Johannessen and Lee, 1974; Molcard and Williams, 1975; Molcard and Tait, 1977; Zodiatis and Gasparini, 1996; Falco et al., 2016; Durante et al., 2019; and Taillandier et al., 2020; and more). It would be advantageous to highlight this consistency with existing literature, or alternatively, to include this information in the Introduction section.

Thanks. Information on the presence and permanence of staircases in the Tyrrhenian Sea and references have been included in the introduction.

• Additionally, please consider incorporating the graphical positions of the two eddies in Figure 8 and adding titles to the maps, such as 'Cruise' and 'Floats.'

We added the labels 'Cruise' and 'Float' to distinguish the two sets of data. We have omitted to add the position of the eddies to avoid making the figure too complex.

**Discussion 4.1**

The content in paragraphs 382–393 contains qualitative speculation and should be rewritten, as it lacks sufficient adherence to the presented results.

It is important to distinguish between what constitutes a measured result (and where to locate it in the previous text) and what represents a conjecture. While I understand this paragraph relates to Results Section 3.2.2, both sections require restructuring and more substantive argumentation to enhance coherence. Consider these questions: Can you demonstrate a definitive influence of the eddies on staircase structures, or is it merely a qualitative correlation? Did you select station 20 due to its proximity to the 12 E° eddy, with the intention of

investigating the relationship between the eddy and the potential development of steps? If this is the case, please state it explicitly to clarify your speculative assertions regarding their interaction.

Additionally, you mention that vertical currents behave as expected (line 270). In what manner can they visibly alter the formation of staircases (line 387), particularly since there are no staircases present at the station nearest the eddy? Can you compare your cast in station 20 with other previous data of the same station? A potential approach could involve starting from the persistence of the center-basin staircases and hypothesizing that the absence of staircases at your station 20 (and possibly adjacent stations, are you able to show them for comparison?) may also be attributable to the presence of the eddies and their associated stronger currents. This interpretation reflects my understanding of this segment of the discussion.

For your reference, a relevant study examining the relationship between a Meddy and staircases is presented in Hebert (1988).

We chose Station 20 to study the relationship between the eddy and the potential development of staircases. It is one of the few deep stations, along with shallow stations near the basin boundaries, where there are no staircases above 1000 m. This station is in close proximity to the 12° E eddy.

We agree that this statement is based solely on the co-occurrence of the presence of the eddy and the absence of thermohaline staircases, and that there is not necessarily a causal relationship. However, we could argue for this effect based on the results recently published by Yang et al. (2024), who showed that the presence of thermohaline staircases in the Caribbean Sea is often perturbed by subsurface mesoscale eddies. We have rephrased our discussion on this topic and only suggested possible interactions between thermohaline staircases and oceanic processes based on this work.

In this section we focus mainly on the spatial and temporal stability of thermohaline staircases in the central part of the basin (according to our observations and the literature); a stability that is likely to have a notable effect on the settling and transformation of particulate and dissolved material, which we detail in the following sections.

Yang, S., Zhang, K., Song, H. et al. Disruptions in thermohaline staircases caused by subsurface mesoscale eddies in the eastern Caribbean Sea. Nature Communications Earth & Environment 5, 408 (2024). https://doi.org/10.1038/s43247-024-01577-3

**Discussion 4.2**

Please consider including a description of the size classes presented in either the text or a table for clarity. A summary of the particle abundance, categorised by size class, from the UVP measurements at all sampling stations is given in Table 1 in the supplementary material.

**Discussion 4.3**

Lines 488–489: Since you are comparing your results with those of Taillandier et al. (2020), it would be beneficial to also reference Durante et al. (2021) in this paragraph. The study analyzed heat and salt fluxes of staircases in a nearby area and over a broader portion of the water column, yet it found the strongest thermohaline fluxes occurring within a similar depth range (700 m – 1600 m) as the one you selected. Thank you for suggesting this work of Durante et al. (2021), which we have now included in the comparison with the study conducted by Taillandier et al. (2020). These results confirm the influence zone of the staircases from a physical perspective and therefore significantly strengthen our approach.

**Technical Comments**

Please refer to the attached paper. Consider these comments as a complement to Reviewer #1's feedback.

**Responses to comments from Reviewer #2**

**Major comments:**

I was quite enthusiast reading the abstract. After reviewing the manuscript, I liked the part about the influence of the eddy on the stratification of the Tyrrhenian Sea. However, I'm disappointed by the main focus of the study, which concerns the concept of particle retention by the occurrence of staircases. I find the figures associated with station #9 are unconvincing and inconclusive. In addition, the authors build some interpretations on the sole basis of 4 steps of a single profile (steps of sta. #9), while a significant number of other stations of that same cruise exhibit staircases. Not much is said about the other profiles, especially in terms of particle retention. Those other profiles should also be used to improve the robustness/representativity of the nitrate fluxes associated with staircases.

Note that I fully agree with the process of retention, decrease of the settling velocity, particle aggregation and consequences for the microbial community when there are strong density gradients that act more or less as a barrier: this is physical, no problem.

My criticism is that there is a lack of clear evidence of a retention and associated increase in mineralization on the sole profile presented in this study (#9 with staircases). It seems to me that there is no clear perturbation of the large scale gradients (AOU, nitrate) that are shown on Figs. 7 (and 6) for instance. At the scale of the steps of the staircases, we find the same gradient signs that are simply enhanced since a step 'connects' two homogeneous regions that have been mixed by the salt-fingering process.

I expected the study to exhibit higher concentration of small scale particles, an increase in nitrate, a decrease in AOU at the base of a mixed region just above a step, or, in a step. I see no such significant anomalies, just local gradients at steps that follow the sign of the large scale gradient, with a larger amplitude induced by the adjacent convectively mixed regions, in the same way that salinity and temperature gradients are very locally enhanced between these mixed regions.

We agree that a demonstration based on a single profile could be misleading, or in any case insufficiently conclusive. In the first version submitted, we chose to limit the figures to 2 stations considered to be the most representative, and this was undoubtedly not sufficient. Given the resolution limitations of the sensors and the variability of the environment, it is often difficult to clearly identify anomalies in particle, nitrate and oxygen concentrations that reflect retention and mineralisation at density interfaces. It's also clear that there must be a time delay in these phenomena. The intensity of mineralization (nitrate production and oxygen consumption) varies over time and must result in a patchy distribution that is impossible to capture at current observation scales.

We now present the complete profiles for two stations (station 20 without significant staircases and station 9 with marked staircases) and close-ups of the staircases of three different stations (stations 8, 9 and 11). We have added the numbers of these stations to the sections and have provided a brief explanation of their selection in a new section at the end of the material and method section (2.3 Selection of representative stations). We believe that the impact of these physical barriers on biological activity is made more credible by the addition of these profiles.

**Further comments are provided below.**

**Detailed comments:**

Fig. 4: The down- and up-casts estimates of vertical velocity are remarkably consistent on profile a, and somewhat less on profile b between 150–250 m and 500-1100 m. Is it physical and in that case what is the source of the variability?

The upcast typically provides lower quality data than the downcast, due to the wake effect of the ascending CTD package disturbing the water column, as well as potential temporal changes in ocean conditions (internal mixing, energy dissipation of currents) between the two casts.

We have simplified this section by presenting figure 4, which shows a histogram of the vertical velocities, and by pointing out in the text the range and standard deviation of vertical velocities.

I. 282: '... the downward movement of water around the anticyclonic eddy': interesting! Can this be also evidenced on a 2D-plot transect of the vertical velocity (as a supplementary subplot of Fig. 3)? Possibly also visible on theta plot Fig. 2A, no?

Anticyclonic eddies play a significant role in the subduction and transport of water masses. Planktonic particles and biomass in anticyclonic eddies tend to follow isopycnal surfaces, resulting in vertical displacement (Rubio et al., 2005. A field study of the behaviour of an anticyclonic eddy on the Catalan continental shelf – NW Mediterranean – . Progress in Oceanography, 66, 2–4, 142–156; Samuelsen et al., 2012. Particle aggregation at the edges of anticyclonic eddies and implications for distribution of biomass. Ocean Science, 8, 389–400). However, this is not the subject of this work.

I. 288–291 + Fig. 5c: we rather observe a mimima between 200–300 m, no? The maxima are below. Is the zonal fragmentation of the mimina 'tongue' between 200–300 m caused by the upward motion of the large reflectors at night? Maybe a slight reformulation is needed to clarify.

The fragmentation of the backscatter index into 'tongues' of lower intensity between 200–300 m is related to diel vertical migration (also known as upwelling) of some organisms during the night.

I. 292 and Fig. 5d: I'm not an expert in this field. A short explanation of the reasons we expect such a distribution would be welcome. Thanks!

The distribution of particles suspended in the ocean is often characterised using the Junge index (denoted  $\alpha$ ), which fits a power law to the particle size distribution, with steeper slopes (higher  $\alpha$ ) indicating a dominance of smaller particles over coarser particles. The higher Junge index observed at depths of approximately 200 to 1000 metres on the section indicates that the pool of particles present in the water is dominated by smaller particles. This predominance decreases at greater depths.

I. 384–385: 'the presence of significant staircase structures down to 2000 m can also be influenced by mixing induced by cyclonic eddies...': Is it the eddy that breaks the staircase structure that was in place, or the fact that the eddy was formed in a region without any staircase structure that is later advected in the middle of the Basin? Do we know where it was generated?

We believe that the eddy either broke the structure of the staircase that was in place, or prevented the formation of stairs.

I. 399: 'a significant reduction in the abundance of fine particles as seen by transmissometry (BAC) under each interface': The decrease is across the interface, not under (under = the mixed layer)

\*\*Right. Corrected\*\*

I. 401 and following, Fig. 9–10 and observations: If I understand, we are looking at the variation across a step (=interface). If I look at the difference in abundance between just above a step minus just below the step, there is no clear rule, even for the two smallest size ranges on Fig. 9; for the two smallest sizes <128 μm: first step = increase with depth, second step = decrease with depth, third step = increase, for Fig.10: decrease with depth. I'm a little puzzled by the concept of 'retention' of small particles when 'simply' looking at Figs 9 and 10. Given the terminology used here, I would have expected to see a peak of small particle abundance within a step or immediately above the step. This does not happen. The overall abundance profile decreases with depth in the absence of staircases (Fig. 5 sta #20). In the step region, this overall profile is mixed in the convective regions associated with the salt finger. A step is not associated with an increase (retention) of small particles, it is just a strong gradient connecting two convectively mixed regions (just as the temperature over a step shows larger local gradients than it would if there were no adjacent mixed layers due to the double diffusion process)... Am I wrong in thinking that if there is retention, the time for small particles to aggregate and possibly become heavier, I should then see an increase in the abundance of small particles at the base of a mixed region above a step?

You are right, the increase in the abundance of small particles on the stairs was not clear on the profiles presented previously, but as we explained above, the option was to present a limited number of figures. We recognise that this was unwise and have now added close-ups of the various parameters at the level of density interfaces for different stations. But it should not be forgotten that this is a variable environment, and that the scales involved are difficult to grasp with current measuring equipment.

I. 447: 'The vertical velocities of the current ... likely to alter the settling of particles': not sure to correctly understand the idea. You mean that a layer that is connectively mixing has down- and up-ward currents at very small scale, what can homogenize the distribution of small scale (almost neutrally buoyant) particles, preventing their deposit at the base of the convective region??

This is indeed what we wanted to express. Neutrally buoyant particles are likely to be homogenized with the mixed layer between two density interfaces.

I. 459 and following: OK for mineralization process, consumption of oxygen, release of nutrients. So what do we expect? Increase in AOU just above a step, since a step acts as a barrier, or across a step, the time for small particles to cross it at a reduced downward velocity? It is not clear from what is said on the process of increase mineralization.

We have reformulated and detailed this part of the discussion on the impact of staircases on biogeochemistry: 'The presence of thermohaline staircases appears to affect the sedimentation pattern of particles; promoting particle retention and aggregation, increasing the time particles spend in the water column. These structures also create distinct microenvironments with gradients of oxygen and nutrients, which are believed to be the result of the degradation of particulate organic. This effect is illustrated for three separate interfaces in figures 9 to 11. It can be then hypothesized that the increase in particle residence time favours the remineralization process, which releases nutrients while consuming dissolved oxygen through heterotrophic respiration. Depending on the conditions and with a sufficiently long stability period, these activities can lead to localized oxygen depletion and an increase in AOU. In our data, this effect is clearly observed in the shallower staircases, between 750 and 900 m, at station 09 (Fig. 11, bottom row). For deeper interfaces for stations 08 and 11, at around 1200 m (Figs. 9 and 10, bottom row), the increase in AOU and the potential accumulation of nitrate are masked by a stronger vertical oxygen gradient and higher nutrient concentrations compared to shallower depths. Moreover, biological activity within these deeper staircases is also reduced due to the decreased lability of organic matter with increasing depth (Ghiglione et al., 2009; Karl et al., 1988). The persistence of these staircase structures allows sufficient time for significant biogeochemical transformations to occur. Arístequi et al. (2009) and Nagata and Kirchman (1997) showed that in deep marine environments, the degradation kinetics of organic matter, which vary according to temperature, oxygen availability, and microbial composition, can range from a few days to several months. This can lead to the local accumulation of nitrates, creating pockets of higher concentration and intensifying vertical double diffusive nitrate fluxes toward upper layers. However, observing these accumulations with the present resolution and

Arístegui, J., Gasol, J. M., Duarte, C.M., Herndld, G. J., 2009. Microbial oceanography of the dark ocean's pelagic realm. Limnology & Oceanography 54, 1501–1529.

Baumas C, Bizic M (2024) A focus on different types of organic matter particles and their significance in the open ocean carbon cycle, Progress in Oceanography, 224, 103,233,

Ghiglione, J.-F., Conan, P., Pujo-Pay, M., 2009. Diversity of total and active free-living vs. particle-attached bacteria in the euphotic zone of the NW Mediterranean Sea. FEMS Microbiology Letters 299, 9–21.

precision remains challenging.'

Karl, D.M., Knauer, G. A., Martin, J. H., 1988. Downward flux of particulate organic matter in the ocean: a particle decomposition paradox. Nature 332, 438–441.

Kiorboe T (2001) Formation and fate of marine snow: small-scale processes with large-scale implications. Sci. Mar. 65(2): 57–71

Nagata, T., Kirchman, D. L., 1997. Roles of Submicron Particles and Colloids in Microbial Food Webs and Biogeochemical Cycles within Marine Environments, in: Advances in Microbial Ecology. Springer, Boston, MA

Thiele S, Fuchs BM, Amann R, Iversen MH 2015. Colonization in the Photic Zone and Subsequent Changes during Sinking Determine Bacterial Community Composition in Marine Snow. Appl Environ Microbiol 81,

For the first two upper steps (Fig. 11), there is an increase in AOU across the step, for the third step around 875 m, the AOU is constant or slightly decreasing, and for the fourth large step the AOU strongly decreases across the step. For the nitrate, there is some increase across the upper three steps but the large scale gradient shows an increase with depth; therefore, it is difficult to conclude that the increase in nitrate across the steps is associated with an increased mineralisation process due to particle retention by just looking at those figures. The difficulty may be due to the presence of large scale gradients, that may mask the signal as described al l. 463. Looking at other profiles with staircases may provide more convincing evidences.

You are right, which is why we added close-ups of the various parameters at the level of density interfaces for different stations as suggested.

**Minor comments:**

Fig. 1a: add the ship route on the map. According to Fig. 3b, it seems that there is an interruption of the zonal 'linear' progression of the ship around the eastern edge of the anticyclonic eddy...

Corrected. We have added the ship's route and explained the brief deviation to the north (recovery of the profiling float) on the station map in Figure 1.

For all transect figures (Fig. 2, 3c, 5, 8): add the station number on the upper abscissa.

We have added the station numbers to the station map in Figure 1 and also along the upper abscissa of the sections.

l. 254 the typical maxima looks like more 10–15 cm/s on Figs 3bc, rather than 30 cm/s. Maybe the exceptional value of 30 cm/s is very locally reached along a slope, but this is not evidenced on Fig. 3.

The values of 30 cm/s are indeed surface speeds observed both by ADCP measurements and derived from altimetry measurements at the level of the eddies. We have corrected the text to indicate representative speeds values.

Fig 3a: give the meaning of the arrows and provide their scale and the source of the data (gridded geostrophic currents from...) + plot the ship route on this map

The arrows indicate the direction and magnitude of the surface geostrophic currents. The origin of the product is now indicated in the figure caption. A scale has been added.

I. 341: figure referencing: Fig. 8 instead of Fig. 6? *Indeed. Corrected.*

I. 344: repetition: 'Staircases are absent under the deepest eddy (about 12° E)', already said in the following sentence (I. 345)

Corrected. The paragraph has been reworded.

I. 397: interface/step starts around 1250 m and ends near 1300 m. To avoid confusion in the wording, you may choose another term than interface if you depict a region encompassing mixed layers and steps.

We have modified the text to indicate the average depth of the interfaces for the selected stations we have chosen to display.

I. 427: unit: 'µm' instead of 'm' *Corrected.*

I. 491: 'stronger than' instead of stronger that' Corrected.

l. 510: extra dot.. *Corrected.*

Citation: https://doi.org/10.5194/egusphere-2024-3436-RC4

---

## Referee Report (RR1)

Title: Effect of double diffusion processes in the deep ocean on the distribution and dynamics of particulate and dissolved matter: a case study in Tyrrhenian Sea

Author(s): Xavier Durrieu de Madron et al.

MS No.: egusphere-2024-3436 MS type: Research article

Iteration: revised, submission #2

First, I would like to thank the authors for their detailed responses and for clarifying several points. The manuscript has been improved in several ways. For the most important: inclusion of additional examples of the influence of staircases on the particle retention process from profiles other than the initial two, better description and explanation of some of the processes with new relevant references, improved part 4.3 with a detailed view of the diffusive fluxes over the water column at station 9 instead of just one depth level.

My main point in the first review was the lack of strong evidence of the retention process of particles by staircases. The additional examples (Figs. 9 - 11) provide close-up views of the details at three steps on profiles that were not shown before. Does the revised version convince me? Unfortunately, I would say, not really, and I will explain why below.

I do understand the comments that measurements at meter scale (the scale of the steps/interfaces) of BAC, large particle abundances, ... are not easy at all, and that there is a strong natural variability at small scales, that sensors have limited response time and noise, ...

Several times, I have tried to convince myself that, what I observe (with the help of your text) in the figures clearly shows that staircases alter the "normal" (without staircases) behavior of particle settling. It's clear that the profiles are different since one has a sequence of steps and layers in your data (even if it's a bit noisier for some variables), the other shows a more usual gradual vertical gradients, as you mention. But my key question is: do the data show an anomaly in particle distribution, nutrient production or AOU, when there are staircases compared to when there are no staircases? And I'm still puzzled when I try to reconcile what I see in the data with your claim that clear evidences exist for particle retention process by staircases.

For example, the discussion of the two large-scale profiles (#9 and 20) falls short of the argument of an aggregation-retention process, which is the main focus of your paper. I don't know what to conclude at the end of line 356. Yes, staircases are also visible for oxygen, nitrate, small particles concentrations, and somewhat less so for large particle abundance. You end the short section with a comment about the Junge index decreasing more in the case of staircases, but, how should I interpret that?

Conversely, I could argue that the concentration of small particles decreases more when there are no staircases (no stairs. :0.447  $\rightarrow$  0.440, with stairs: 0.448  $\rightarrow$  0.4435). I would expect the opposite if steps act as a "barrier" with aggregation of fine particles. Moreover, I could also argue that there is about the same abundance of large particles between the two profiles at 2000 m, that is after having crossed all the interfaces of the staircases (i.e. no excess of the large particle production when there are staircases, as I would expect with retention-aggregation). The large particle profiles are difficult to compare, because at the origin, at 600 m, the abundance is not the same. The profiles do not start with the same 'initial' conditions. Therefore, what I just wrote 2 lines above about the decrease in large particles is biased, as well as what you wrote on lines 355-356: it is difficult to interpret the decrease rates in large particles with depth since the starting points are not the same (for large particles, and thus, for the

Junge index). The proximity of the anticyclone may bias the no-staircase profile as marine production in the euphotic zone strongly depends on the vorticity field (e.g. Belkin et al. 2022, 10.5194/os-18-693-2022). This adds another degree of complexity to the staircases-nostaircase comparison.

Fine particles: should decrease more with staircases (consumed by aggregation)
Large particles: should decrease less with staircases (produced by aggregation)
Thus Jungle index (fine/large) should decrease more with stairs, but the starting point in terms of fine and large particle abundance should be the same at shallow depth (above the first step of the staircase structure) for the comparison to be meaningful.

Finally, we are left with the possible impact of staircases with the three close-up examples of steps (Figs. 9-11). The discussion of these figures ends up on l. 486-487, with the statement that the observed variations provide 'strong evidence' that staircases have a significant influence on particle abundance and size distribution. I agree that these figures show that staircases alter the distribution of the finest particles seen by the transmissiometer, the oxygen and the nitrate concentrations. This is easily seen when I compare the interface gradient with the upper and lower mixed layer from off the interface in these examples. For large particles (UVP measurements by particle size), there is not much evidence, point-to-point measurements show a large noise, noise that is as large as what the authors are trying to estimate. And therefore, I do not agree with the statement: l. 481, 'an increase is evident for the largest particles' (largest class seen by the UVP). Then follows a useful paragraph on the retentionaggregation process with references.

My conclusion is that the close-up views on these three interfaces do not show any clear evidence of retention-aggregation process, the main topic of the paper according to the abstract. They only show that mixed layers homogenize the distribution of fine particles (and nitrate, O2), and that in between, there are more or less large interfaces connecting the two mixed layers, as one would expect if one creates two successive mixed layers from an initially smooth constant gradient of properties. Unfortunately, this has nothing to do with proving the existence of a retention-aggregation process in this case of small density gradients across the interfaces.

To get such a proof, you have to show that staircases remove more fine particles and produce more large particles than you would observe in a region without staircases. This is very hard to do as the real ocean is not a controlled lab experiment where the initial conditions are the same. Given the large variability in your examples of interfaces, I would suggest doing a statistical study among all interfaces of the staircases observed in your entire data set (not just a pick-up of very few examples). For example, you could statistically study what happens within mixed layers on the one hand, and across interfaces on the other hand (sign, intensity of gradients, variability inside the mixed layer and inside interfaces), try to relate things to some parameters (dependence on the amplitude of the interface density gradient, ...) and try to contrast these statistics with what is observed for the set of profiles without staircases and for the same depth ranges. You might find some undeniable statistical evidence of a retention-aggregation process. You could also look at the whole staircase structure of each profile, and try to relate whole depth variations in the abundance of fine and large particles, to the number of interfaces in the profiles, the cumulated thickness of interfaces, the intensity of the sum of the density gradients, ... compare that with the 'typical' variations observed when there is no staircase and see if something clear emerges.... From my point of view and given the oceanic natural and instrumental variability, you cannot avoid such a statistical study to support unquestionably your conclusions about the aggregation-retention process, the main subject of your study.

**Other comments:**

- l. 261 269: water masses acronyms have already been defined on page 3. No need to repeat.
- Fig. 2: add TIW on the subplots
- 1.276: 'The anticlyclonic eddy...': first time it is mentioned, change for 'An anticlyclonic eddy...' 1.277 280: You can also mention the signature of the cyclonic eddy to the west of the anitcylone on the property distribution.
- l. 288: 'The cyclonic eddy...', to be more accurate, I 'd write 'The southern edge of the cyclonic eddy' given Fig. 3a. The center is sensibly further North.
- Fig. 3a: add station numbers
- Fig. 4. To me, not essential. Could be removed if you need room for other figures.
- 1. 317: To better locate immediately the turbid tongue, add : (sta. #19 20).
- l. 424 and following: In your data, staircases largely disappear under the anticyclone, not just on the periphery (although the section does not cross the core). There is no such removal of staircases in the vicinity of the cyclone to the west. Another possibility for eroding staircases, with a contrasting behavior between cyclones and anticyclones, comes from the focusing of near inertial internal waves and associated enhanced velocity shear and mixing in anticyclones (e.g. Fer et al., 2018: The dissipation of kinetic energy in the Lofoten Basin Eddy, J. Phys. Ocean. 48, pp. 1299-1316. Doi:10.1175/JPO-D-17-0244.1).
- Section 4.2, l. 451-455: This is the first time you mention the particle size spectrum and its slope (?), while no such spectrum is shown in the study. It's a bit confusing. I guess you are simply referring to the Junge index that was discussed before, which is a very rough proxy of what a real spectrum would be. I don't think the wording 'spectrum' has to be used in this study. The whole discussion can be done using the simpler Junge index that you show and discuss.
- l. 454: 'deep scattering layer', why scattering? Meaning in this context?
- l. 468 469: 'macroscale' and 'microscale' are unfortunate in this context. Usually, in physical oceanography, vertical large-scale = O(>100 m), finescale (1 m 10 m), microscale (1 cm 10 cm).
- 1. 474: the largest thickness among the three examples reaches ~150 m.
- Fig. 9 11: indicate what is the red line (mean, median, ...)
- 1. 578 579 : homogenize writing of unit exponents
- Fig. 12: Thanks for this useful figure. Is the divergence in nitrate fluxes between two interfaces compensated for by nitrate production? If yes, does the number make sense in terms of nitrate production by processes?

---

## Referee Report (RR2)

**Second review of EGUsphere-2024-3436**

Effect of double diffusion processes in the deep ocean on the distribution and dynamics of particulate and dissolved matter: a case study in Tyrrhenian Sea by Durrieu de Madron et al.

**Overview**

The manuscript by Durrieu de Madron et al. presents CTD, ADCP and optical profile data along a section crossing the Tyrrhenian Sea and investigates salt finger staircase effects on particulate and dissolved matter distribution.

The authors have significantly improved the readability and presentation of their results, but there are still a few corrections and clarifications to be made, and some points that could be improved. Therefore, I recommend a minor revision based on the comments below.

**1 General comments**

a) Add Richardson number Ri to Fig. 3, and buoyancy frequency  $N^2$ , vertical shear  $S^2$ , Ri depth profiles in Figs. 6 and 7; expand the discussion on turbulent mixing in terms of shear instability to support your claims on lines 414-443.

**2 Specific comments**

Line 167: It is hereafter referred to in the

Line 269: visible below

Fig. 2: I would have expected distance to increase with increasing station number, so this is a bit confusing to me. Maybe at least mark east and west? Also, the panels are not aligned.

Fig. 2 caption: Upper x axes show ...

Line 286: Dynamic topography and observed horizontal velocity reveal

Line 295: Figure 3: (a) Absolute dynamic topography and surface geostrophic currents derived from daily Copernicus Marine Environment Monitoring Service Gridded L4 data (interpolated to 3.75 arcmin), averaged over the period of the cruise (14–16 March 2020). (b)

Line 301: between 21

Line 305: upper 300 m. Also, what do you mean by vertical shear,  $\partial w/\partial z$ ? This is potentially interesting since it represents divergence. Is the sign inside the eddy consistent with what you

expect from isopycnal slope?

Fig. 5: mark (a), (b) etc

Line 327: tongues?

Line 359, 367: delete 'and'

Fig. 7 caption: 'Major density steps are delineated by horizontal grey lines. The horizontal dashed lines indicate the base of the main density interfaces.' - these two sentences are redundant

Line 373: position

Line 393: (a) Depth

Line 394: basin measured by

Fig. 8 is inconsistent between (a) and (b). Firstly, (a) is not marked. I presume the dots in (b) are steps identified using the procedure referenced in Methods; dots are lacking from (a). (a) shows isopycnals, (b) does not. What is 'average layer depth' in y axis label in (b), is it just depth (the float was profiling, not in isopycnal following mode, correct?)? Please present consistent information between the panels and clearly explain what is illustrated in the caption.

Line 404: wind forcing

Line 427: of concurrent direct

Line 429: delete 'This observation is consistent with the emerging understanding of the dynamic interplay between mesoscale features and fine-scale thermohaline structures in the ocean interior.' - redundant

Line 434: highlighted the persistence of

Line 467: in suspended

Line 468: coarse particle fraction

Line 468-469: 'microscale' in the ocean is O(mm)-few cm! And 'macroscale' is not established nomenclature for a specific vertical scale. Please replace 'macroscale' and 'microscale' by something like 'scales between / of the order of ...' to be specific.

Line 506: delete 'in the literature'

Line 515: interfaces

Line 542: sedimentation patterns, promoting

Line 546: organic matter?

Line 572: delete 'in'

Line 577: assessed at the depth of each step

---

## Referee Report (RR3)

Title: Effect of double diffusion processes in the deep ocean on the distribution and dynamics of particulate and dissolved matter: a case study in Tyrrhenian Sea

Author(s): Xavier Durrieu de Madron et al.

MS No.: egusphere-2024-3436 MS type: Research article

Iteration: revised, submission #3

My main comment on the second revised version concerned the lack of clear evidence of retention-aggregation process. To address this point, the authors have now included two new plots which make use of a larger number of CTD profiles to statistically show the influence of staircases on the Junge index and the AOU. In my view, this is a useful addition that provides clearer evidence of the retention-aggregation process. While the differences between the two groups of profiles (with and without staircases) in the decreasing slopes of the mean Junge index and AOU with depth are weak, they exist. This new figure provides more convincing arguments than the former versions of the manuscript, and I thank the authors for this improvement. Statistically speaking, more could probably be done, but as it is now, the manuscript has already gained in strength, and the arguments are more convincing to readers.

I don't have any further major comments regarding the core of the manuscript.

However, this new version includes a point about the Richardson number (Ri) in response to a query from one of the other two reviewers. Using this Ri causes confusion and does not provide any useful information:

My first comment relates to the way Ri is calculated, which makes use of buoyancy frequency (from CTD) and shear (from LADCP) profiles, both smoothed over a large scale (48 m, l. 268). At this scale, it is unlikely that shear-driven favourable regions will be detected. This is because, at these scales, the shear is usually far too weak and that open-ocean instabilities far from boundaries occur at meter-scale. The only reason Ri is sometimes below the theoretical value of ¼ in this dataset is due to the very weakly stratified layers. Shear-driven turbulence is driven by shear, not by weak buoyancy!

From my experience, even with a 10-m scale LADCP shear and  $N^2$ , it is difficult to associate Ri (say lower than  $\frac{1}{4}$ , or lower than some number such as 1 to account for the low vertical resolution of the LADCP shear profiles) with, for instance, enhanced levels of turbulent kinetic energy dissipation rates (i.e. measured turbulence levels) unless you sample very active, shear-driven turbulent regions. Thus, using 48-m scale resolution in an open-ocean staircase environment, a worse configuration, will only exhibit weakly stratified regions, which could be achieved using  $N^2$  only. Attempting to link Ri < 1/4 and turbulent mixing is meaningless with this configuration (l. 487 - 497).

My second comment relates to the interpretation of Ri in the context of staircases. As mentioned (l. 532-546) in the text and by previous studies, vertical shear (low Ri) tends to prevent the onset of salt-finger instability. At l. 495 – 497, because staircases have almost homogeneous layers that consequently drop Ri below ¼, the authors state that turbulent mixing ( = shear-driven here) is "concentrated in the homogeneous layers of the staircases". The fact that homogeneous layers exist is not linked with a shear-driven process but with a convective salt-fingering process. Using a 48-m scale Ri cannot be used to infer the existence of enhanced shear-driven turbulence in those homogeneous layers.

l. 566-57: "Below 1000 m, reduced stratification and shear create low Ri conditions, which appear

to be important for both the formation and maintenance of thermohaline staircases, as evidenced by both float measurements and cruise data in the basin's central region"

So you mean that low Ri ( = shear-driven turbulence, if you can ever diagnose it with this Ri) is helpful for the formation and maintenance of thermohaline staircases ?! This contradicts what you wrote about previous studies at l. 540-546. Some clear explanations of the processes are needed.

---

## Referee Report (RR4)

**Third review of EGUsphere-2024-3436 REV#2**

Effect of double diffusion processes in the deep ocean on the distribution and dynamics of particulate and dissolved matter: a case study in Tyrrhenian Sea by *Durrieu de Madron et al.*

The authors have responded well to the reviewers' suggestions, and I believe the overall structure and readability of the paper, including the figures, are now excellent. The addition of stratification and stability parameters is useful, and classifying stations with and without staircases is an effective choice. There is a notable improvement in Figure 8. Significant progress has been made in the statistical analysis, as wisely recommended by

Reviewer #1. Despite this, I would still like to make a few comments:

Paragraph 2.3: The methods of threshold-based detection of stations could be expanded. It's clear how you selected the WStairs stations, which are six, but it's less clear why stations 4, 5, 6, 17, and 18 are excluded from the NoStair category. Why don't they fit your classification criteria? Splitting stations into two groups smooths out local variability, but it also assumes that the groups are comparable within each category. Heterogeneities within groups could affect the slopes and the significance, that's why you are not using all the 16 stations? This should be clarified.

Line 422: There's a misalignment issue in Figure 5 with the numbers in the subplot.

Figure 9: When you mention shifted AOU profiles, do you mean both graphs start at 700 m, even though the linear regression for AOU begins at 1000 m? If so, the last two lines of the caption may be more confusing than helpful, as the graph is self-explanatory. Additionally, depth is plotted on the y-axis in all other figures and plots, including those for individual stations... except in this one. If this does not affect the readability of the plot from a statistical perspective, you might consider flipping the axis.

Line 612: You define WStairs and NoStair in 2.3, and paragraph 3.2.2 is about the positioning and persistence of the main staircases.

Line 614: If I am not mistaken, Aiken et al., 1991 refers to a book. Could you specify the exact location of this reference within your manuscript?

Line 610- 625: While the slopes are very close in value, you report a p-value indicating that the difference is statistically significant. But, given the small magnitude of these differences

and the brief description of your statistical approach, this section could benefit from a more detailed explanation to clarify the reasoning behind the significance and its implications. For instance, you could include some details about the statistical approach in the Methods section (Paragraph 2.3), which would help keep the discussion section more concise and focused without losing information

---

## Referee Report (RR5)

**Third review of EGUsphere-2024-3436**

Effect of double diffusion processes in the deep ocean on the distribution and dynamics of particulate and dissolved matter: a case study in Tyrrhenian Sea by Durrieu de Madron et al.

**Overview**

The manuscript by Durrieu de Madron et al. presents CTD, ADCP and optical profile data along a section crossing the Tyrrhenian Sea and investigates salt finger staircase effects on particulate and dissolved matter distribution.

The authors have adequately addressed all the reviewers' comments which has strengthened the manuscript. I only have some minor further comments below.

Line 308: stratification is lower

Figure 3d: Ri would be best shown in log10 scale to emphasize Rii0.25

Lines 320-329: The part about vertical velocity seems a bit half-baked, if you are showing the figure perhaps there can be a bit more to say? For example, is there a significant difference in the histogram for w within the steps vs outside of the steps? Or at least between profiles with vs without steps?

Line 322: in the upper 300 m

Line 428: across the basin from west to east during

Line 461: absent in the upper 1000 m at the stations within the anticyclonic eddy

Line 473: delete 'drops'

Line 613: particulate organic matter? .

Line 653: vertical flux

Line 667: vertical fluxes

---

## Author Response (AR2)

**List of all relevant changes made in the manuscript**

We addressed the main concern about the lack of strong quantitative evidence for particle retention by thermohaline staircases by applying a statistical approach. By analyzing vertical changes in the Junge index across all deep stations and grouping profiles by the presence (termed « WStairs ») or absence (termed « NoStair ») of staircases, we used multiple linear regression to show that the Junge index decreases more slowly where staircases occur. This supports the idea that staircases help to retain larger particles deeper in the water column, and we have included this analysis in the discussion. A similar approach was used with the Apparent Oxygen Utilization that indicated increased in situ mineralization activity for the deep stations having multiple sharp staircases

We improve the labelling of the figures and completed their captions as requested. We incorporated information on the Richardson number and shear turbulence and added the relevant variables for the two representative stations 09 (« WStairs ») and 20 (« NoStair »). We used these results to detail the effect of turbulence in terms of shear instabilities for different types of profiles in section "4.1 Circulation and thermohaline staircases" of the discussion. This part of the discussion has been substantially reworded.

We clarified the terminology (e.g., Junge index as proxy for particle size distribution, deep scattering layer" as micronekton detected by ADCPs) and replaced ambiguous terms throughout the text.

**Point-by-point response to the reviews**

Second review of EGUsphere-2024-3436

Effect of double diffusion processes in the deep ocean on the distribution and dynamics of particulate and dissolved matter: a case study in Tyrrhenian Sea by *Durrieu de Madron et al*.

First, I would like to thank the authors for their detailed responses and for clarifying several points. The manuscript has been improved in several ways. For the most important: inclusion of additional examples of the influence of staircases on the particle retention process from profiles other than the initial two, better description and explanation of some of the processes with new relevant references, improved part 4.3 with a detailed view of the diffusive fluxes over the water column at station 9 instead of just one depth level.

My main point in the first review was the lack of strong evidence of the retention process of particles by staircases. The additional examples (Figs. 9 - 11) provide close-up views of the details at three steps on profiles that were not shown before. Does the revised version convince me? Unfortunately, I would say, not really, and I will explain why below.

I do understand the comments that measurements at meter scale (the scale of the steps/interfaces) of BAC, large particle abundances, ... are not easy at all, and that there is a strong natural variability at small scales, that sensors have limited response time and noise, ...

Several times, I have tried to convince myself that, what I observe (with the help of your text) in the figures clearly shows that staircases alter the "normal" (without staircases) behavior of particle settling. It's clear that the profiles are different since one has a sequence of steps and layers in your data (even if it's a bit noisier for some variables), the other shows a more usual gradual vertical gradients, as you mention. But my key question is: do the data show an anomaly

in particle distribution, nutrient production or AOU, when there are staircases compared to when there are no staircases? And I'm still puzzled when I try to reconcile what I see in the data with your claim that clear evidences exist for particle retention process by staircases.

For example, the discussion of the two large-scale profiles (#9 and 20) falls short of the argument of an aggregation-retention process, which is the main focus of your paper. I don't know what to conclude at the end of line 356. Yes, staircases are also visible for oxygen, nitrate, small particles concentrations, and somewhat less so for large particle abundance. You end the short section with a comment about the Junge index decreasing more in the case of staircases, but, how should I interpret that?

Conversely, I could argue that the concentration of small particles decreases more when there are no staircases (no stairs.  $:0.447 \rightarrow 0.440$ , with stairs:  $0.448 \rightarrow 0.4435$ ). I would expect the opposite if steps act as a "barrier" with aggregation of fine particles. Moreover, I could also argue that there is about the same abundance of large particles between the two profiles at 2000 m, that is after having crossed all the interfaces of the staircases (i.e. no excess of the large particle production when there are staircases, as I would expect with retention-aggregation). The large particle profiles are difficult to compare, because at the origin, at 600 m, the abundance is not the same. The profiles do not start with the same 'initial' conditions. Therefore, what I just wrote 2 lines above about the decrease in large particles is biased, as well as what you wrote on lines 355-356: it is difficult to interpret the decrease rates in large particles with depth since the starting points are not the same (for large particles, and thus, for the Junge index). The proximity of the anticyclone may bias the no-staircase profile as marine production in the euphotic zone strongly depends on the vorticity field (e.g. Belkin et al. 2022, 10.5194/os-18-693- 2022). This adds another degree of complexity to the staircases-nostaircase comparison.

Fine particles: should decrease more with staircases (consumed by aggregation) Large particles: should decrease less with staircases (produced by aggregation) Thus Jungle index (fine/large) should decrease more with stairs, but the starting point in terms of fine and large particle abundance should be the same at shallow depth (above the first step of the staircase structure) for the comparison to be meaningful. Finally, we are left with the possible impact of staircases with the three close-up examples of steps (Figs. 9 - 11). The discussion of these figures ends up on 1. 486 – 487, with the statement that the observed variations provide 'strong evidence' that staircases have a significant influence on particle abundance and size distribution. I agree that these figures show that staircases alter the distribution of the finest particles seen by the transmissiometer, the oxygen and the nitrate concentrations. This is easily seen when I compare the interface gradient with the upper and lower mixed layer from off the interface in these examples. For large particles (UVP measurements by particle size), there is not much evidence, point-to-point measurements show a large noise, noise that is as large as what the authors are trying to estimate. And therefore, I do not agree with the statement: 1. 481, 'an increase is evident for the largest particles' (largest class seen by the UVP). Then follows a useful paragraph on the retention-aggregation process with references.

My conclusion is that the close-up views on these three interfaces do not show any clear evidence of retention-aggregation process, the main topic of the paper according to the abstract. They only show that mixed layers homogenize the distribution of fine particles (and nitrate, O2), and that in between, there are more or less large interfaces connecting the two mixed layers, as one would expect if one creates two successive mixed layers from an initially smooth constant gradient of properties. Unfortunately, this has nothing to do with proving the existence of a retention-aggregation process in this case of small density gradients across the interfaces.

To get such a proof, you have to show that staircases remove more fine particles and produce more large particles than you would observe in a region without staircases. This is very hard to do as the real ocean is not a controlled lab experiment where the initial conditions are the same. Given the large variability in your examples of interfaces, I would suggest doing a statistical study among all interfaces of the staircases observed in your entire data set (not just a pick-up of very few examples). For example, you could statistically study what happens within mixed layers on the one hand, and across interfaces on the other hand (sign, intensity of gradients, variability inside the mixed layer and inside interfaces), try to relate things to some parameters (dependence on the amplitude of the interface density gradient, ...) and try to contrast these statistics with what is observed for the set of profiles without staircases and for the same depth ranges. You might find some undeniable statistical evidence of a retention-aggregation process. You could also look at the whole staircase structure of each profile, and try to relate whole depth variations in the abundance of fine and large particles, to the number of interfaces in the profiles, the cumulated thickness of interfaces, the intensity of the sum of the density gradients, ... compare that with the 'typical' variations observed when there is no staircase and see if something clear emerges.... From my point of view and given the oceanic natural and instrumental variability, you cannot avoid such a statistical study to support unquestionably your conclusions about the aggregation-retention process, the main subject of your study.

**Response to Reviewer 1 – General Comment**

We thank the reviewer for his careful reading and for raising a number of important points regarding the interpretation of particle dynamics in relation to the presence of thermohaline staircases. We fully acknowledge the difficulty of comparing profiles that originate under different initial conditions, and we have taken steps to better quantify the influence of staircases using normalized approaches and statistical testing. By examining the slope of a relevant indicator of the abundance and size of particles over a shared depth range, rather than comparing absolute values, we minimize the effect of these confounding factors.

Therefore, to better understand the link between staircases and particle retention or aggregation, we performed a quantitative comparison of the vertical evolution of the Junge index for all deep stations, which have profiles characterised by multiple significant staircases in some cases and not in others. We selected this index primarily because it captures the relative abundance of fine versus large particles and is particularly well suited to detecting subtle shifts in particle size distributions between 80 and 400  $\mu$ m. Additionally, unlike other parameters, the Junge index exhibited a relative linear trend between 700 and 1900 meters, making it more appropriate for regression analysis. Our analysis based on a multiple linear regression with interaction model, testing the difference in slope between the two groups, shows that :

The Jung index decreases significantly more slowly with depth in profiles containing staircases than in profiles without staircases (slope of  $-0.0007 \text{ m}^{-1} \text{ vs } -0.0008 \text{ m}^{-1}$ ; p < 0.01).

- This result was obtained by pooling data from multiple stations into 2 groups identified as "With Staircases (Wstairs)" and "No Staircase (NoStair)" and fitting independent linear regressions, which reduces the impact of noisy local variations and initial concentration differences.
- The slower decrease in the Junge index suggests that in the presence of staircases, the relative contribution of larger particles to the suspended particle pool is maintained deeper in the water column.

We have revised the discussion (section 4.2) to incorporate these statistical analyses.

We also applied this statistical approach to the AOU for both groups of stations. The results indicate a slower decrease in AOU with depth for stations with multiple staircase steps, which could reflect a more active microbial cycle and a greater contribution of organic matter

degradation to the oxygen balance. We have incorporated this statistical analysis in the discussion (section 4.3).

**Other comments:**

1. 261–269: water masses acronyms have already been defined on page 3. No need to repeat. As you rightly pointed out, the full names of the water masses were redundant at this stage of the manuscript. We have therefore removed them and retained only the acronyms, which had already been introduced earlier – except for TIW, which we now define upon first mention.

**Fig. 2: add TIW on the subplots**

We have added the TIW position to the temperature and salinity subplots, as for the other water masses. The TIW corresponds to the small subsurface water bubble at stations (13, 14, 20, and 19) in the anticyclonic eddy between 11 and 12° E.

1.276: 'The anticcylonic eddy...': first time it is mentioned, change for 'An anticlyclonic eddy...'

Done

1. 277–280: You can also mention the signature of the cyclonic eddy to the west of the anitcylone on the property distribution.

We have modified the sentence accordingly to also mention the cyclonic eddy located west of the anticyclone and its influence on the vertical structure of the water column. The revised sentence now reads:

'An anticyclonic eddy around 12°E and the boundary current jet, especially over the eastern part of the section, modify the intermediate water depth (EIW) by a few hundred metres without significantly affecting the deep water depth (TDW). To the west, the signature of a cyclonic eddy is also visible and contributes to the vertical modulation of isopycnals. These mesoscale structures affect the distribution of oxygen and nitrate, which increase less rapidly with depth beneath them.'

1. 288: 'The cyclonic eddy...', to be more accurate, I 'd write "The southern edge of the cyclonic eddy" given Fig. 3a. The center is sensibly further North.

We agree that the center of the cyclonic eddy is indeed likely located further north, as suggested by Fig. 3a. We have therefore modified the sentence as recommended in the revised manuscript to refer more accurately to "the southern edge of the cyclonic eddy."

**Fig. 3a: add station numbers**

We understand the interest in identifying the station numbers directly on the figure. However, after careful consideration, we chose not to add them in order to preserve the clarity and readability of the figure, which is already quite dense. For consistency and to avoid redundancy, we prefer to refer readers to Figure 1, where the station numbers on the map are clearly indicated. In addition, station numbers are included in the new Figure 3d representing the Richardson number, as requested by the third reviewer.

Fig. 4. To me, not essential. Could be removed if you need room for other figures.

We prefer to keep this figure as the estimation of instantaneous vertical velocities is rarely done and in this case, it is useful to consider their effect on particle sedimentation.

1. 317: To better locate immediately the turbid tongue, add: (sta. #19–20). Done

l. 424 and following: In your data, staircases largely disappear under the anticyclone, not just on the periphery (although the section does not cross the core). There is no such removal of staircases in the vicinity of the cyclone to the west. Another possibility for eroding staircases, with a contrasting behavior between cyclones and anticyclones, comes from the focusing of near inertial internal waves and associated enhanced velocity shear and mixing in anticyclones (e.g. Fer et al., 2018: The dissipation of kinetic energy in the Lofoten Basin Eddy, J. Phys. Ocean. 48, pp. 1299–1316. Doi:10.1175/JPO-D-17-0244.1).

We have modified the text accordingly to reflect this more accurately. We also thank the reviewer for highlighting the potential role of near-inertial internal wave focusing and enhanced shear in anticyclones, as discussed in Fer et al. (2018). We have integrated this reference and the suggested mechanism into the revised version of the manuscript, as it provides a valuable hypothesis to explain the observed erosion of staircases under the anticyclone and the contrasting behavior near the cyclone. We added this sentence:

"This erosion is not observed near the neighbouring cyclonic structure to the west, possibly due to the focusing of near-inertial internal waves and enhanced mixing within anticyclones, as described by Fer et al. (2018) in the Lofoten Basin Eddy."

Section 4.2, l. 451–455: This is the first time you mention the particle size spectrum and its slope (?), while no such spectrum is shown in the study. It's a bit confusing. I guess you are simply referring to the Junge index that was discussed before, which is a very rough proxy of what a real spectrum would be. I don't think the wording 'spectrum' has to be used in this study. The whole discussion can be done using the simpler Junge index that you show and discuss. We agree. We added in the materials and methods section that the Junge index was a proxy for the particle size spectrum, and in section 4.2 we changed the reference to the spectrum to the Junge index. We also changed particle size spectrum by particle size distribution throughout the text.

**1. 454: 'deep scattering layer', why scattering? Meaning in this context?**

The origin of the deep scattering measured from ADCP hull data is explained in section 3.1.3 Turbidity, large particle abundance and Junge index. It essentially corresponds to the presence of live micronekton (small mesopelagic fish, crustaceans and cephalopods). In this part of the discussion on the characteristics of staircase stations compared to 'non-staircase' stations, we want to emphasise that the transition zone below the EIW between 400 and 800 m, where most staircases occur, is also a zone of dominant biological (grazing, predation) and biogeochemical (remineralisation, respiration) activity, which is likely to influence the abundance and nature of particulate material.

l. 468–469: 'macroscale' and 'microscale' are unfortunate in this context. Usually, in physical oceanography, vertical large-scale = O (>100 m), finescale (1 m - 10 m), microscale (1cm - 10 cm).

We have modified this section as follows:

'The overall diminution in suspended particle abundance, concurrent with an augmentation of the coarse particle fraction, is evident not only at large vertical scales throughout the water column beneath the Eastern Intermediate Waters (EIWs), but also at finer scales spanning several decametres within the thermohaline staircases.'

1. 474: the largest thickness among the three examples reaches  $\sim$ 150 m. Done

**Fig. 9–11: indicate what is the red line (mean, median, ...)**

We have now added a description of the data representation in the 3 figures captions to clarify the distinction between individual measurements and averaged profiles. Specifically, we now indicate that 'grey dots represent in situ measurements, while the red line corresponds to the averaged profile.'

**1. 578–579: homogenize writing of unit exponents**

Fig. 12: Thanks for this useful figure. Is the divergence in nitrate fluxes between two interfaces compensated for by nitrate production? If yes, does the number make sense in terms of nitrate production by processes?

Thank you for this question. We have evaluated if the observed divergence in nitrate fluxes between two interfaces could be accounted for by in situ nitrate production via organic matter remineralization. Based on our estimates, the divergence in nitrate fluxes between two successive interfaces ranges from 5 to 10 μmol NO3- m-2 d-1 over a layer thickness of 70 to 250 m.

To assess whether this is consistent with known remineralization processes, we considered published values of nitrate production in the mesopelagic zone (500-2000 m). Reported rates of nitrification and nitrate accumulation due to mineralization vary between:

- 1 to 30  $\mu$ mol NO3- m-3 d-1 in the open ocean (Yool et al., 2007; Clark et al., 2020),
- 5 to 20  $\mu$ mol NO3 m-3 d-1 in the Mediterranean mesopelagic layer (Christaki et al., 2021; Santinelli et al., 2015).

The nitrate production integrated over the water layer thickness is: Production = layer thickness × rates.

- For a layer thickness of 250 m (case for the mixed layer between the interfaces at  $\sim 1000$  and 1250 m), the NO3-production = 250 m  $\times$  5 to 20  $\mu$ mol m-2 d-1 = 1250 to 5000  $\mu$ mol m-2 d-1.
- For a layer thickness of 70 m (case for the mixed layer between the interfaces at  $\sim$ 890 and 960 m), the NO3-production = 70 m × 5 to 20  $\mu$ mol m-2 d-1 = 350 to 1400  $\mu$ mol m-2 d-1.

These values are about two orders of magnitude higher than the observed divergence of 5-10  $\mu$ mol m-2 d-1, indicating that local nitrate production is more than sufficient to compensate for the flux divergence observed across the stair interfaces.

**References**

Yool, A., Martin, A. P., Fernández, C., & Clark, D. R. (2007). The significance of nitrification for oceanic new production. Biogeosciences, 4(4), 447–479. https://doi.org/10.5194/bg-4-447-2007

Clark, D. R., Mayor, D. J., Saunders, R. A., et al. (2020). The seasonal cycle of nitrification in the Northeast Atlantic. Global Biogeochemical Cycles, 34(11), e2020GB006564. https://doi.org/10.1029/2020GB006564

Santinelli, C., Ribera d'Alcalà, M., & Manca, B. B. (2015). The Mediterranean Sea dark inorganic carbon production. Geophysical Research Letters, 42(12), 4752–4760. https://doi.org/10.1002/2015GL064853

Christaki, U., Van Wambeke, F., Lefèvre, D., et al. (2021). Microbial food web functioning in the open Mediterranean Sea: A synthesis of two decades of experimental investigations. Deep-Sea Research Part II, 188–189, 104,939. https://doi.org/10.1016/j.dsr2.2021.10493

**Second review of EGUsphere-2024-3436**

Effect of double diffusion processes in the deep ocean on the distribution and dynamics of particulate and dissolved matter: a case study in Tyrrhenian Sea by Durrieu de Madron et al.

**Suggestions for revision or reasons for rejection**

I have carefully examined the revised manuscript by Durrieu de Madron et al., along with their comprehensive responses to my comments and those of other reviewers. I am pleased to observe a significant enhancement in both the readability of the text and figures, as well as in the scientific content, which is now presented in a more structured and coherent way. In particular, the Introduction and Materials and Methods sections have been significantly improved, they are now comprehensive, complete, and easily understandable. I also greatly appreciate the graphical abstract, which effectively conveys the essence of the article in a single image. Results show now improved figures and comprehensive information. The Discussion section has significantly improved, particularly in how it highlights scientific considerations supported by the results and presents coherent hypotheses and related references to explain the section that was previously regarded as weak. Additionally, the decision to include three more stations near the initially selected ones further enriches the considerations expressed. The conclusions are significant and provide an important element to the understanding of the salt fingers phenomenon, as this study represents the first instance (to my knowledge) where research on Tyrrhenian staircases transitions from abiotic to biotic components. The significance of double diffusion processes is growing in tandem with increased knowledge on the subject, and this study marks a significant step forward in our understanding of how these peculiar dynamics also influence biotic components.

We thank Reviewer 2 for the time and effort dedicated to the review of the revised manuscript.

**Here some minor suggestion:**

Line 46: If salinity is expressed in g kg-1 m-1, you are referencing Absolute Salinity, as recommended by the scientific community. However, since "classical" salinity (PSU) and Absolute Salinity exhibit slight differences in their values (there is a small shift), it would be prudent to use a consistent expression. Given that your study employs salinity (PSU), it is important to avoid potential misunderstandings by clearly specifying when either salinity or Absolute Salinity is cited.

We agree. Although practical and absolute salinities differ by about 0.5% or less, a vertical absolute salinity gradient of 0.5 g/kg/m is almost equivalent to a gradient of 0.5 PSU/m. We have simply changed the unit and expressed the gradient in PSU/m for consistency throughout the text.

Line 184: in the final version there is a blue paragraph here, check it out! Done

Line 143: Fuda et al. (2002) hypothesized that water denser than TDW found in the deep Tyrrhenian (2000–3500 m) might form locally occasionally, but this hypothesis was not supported by further observations. Relevant references can be found in various recent studies, including Durante et al. (2021), which you have already cited. I therefore recommend the removal of this section.

As recommended, we have removed the corresponding sentence and the reference to Fuda et al. (2002).

Line 334: do you mean "potential temperature, salinity etc..."?

We have revised the sentence for improved clarity and readability. It now reads: "In this section we describe the vertical distribution of physical (potential temperature, salinity, potential density anomaly), dissolved and particulate parameters (dissolved oxygen, nitrate, apparent oxygen utilization, beam attenuation coefficient, large particle abundance, and Junge index) at station 20, which shows virtually no significant staircase (Fig. 6), and station 09 which shows marked staircases between 700 and 1700 m (Fig. 7)."

**Second review of EGUsphere-2024-3436**

Effect of double diffusion processes in the deep ocean on the distribution and dynamics of particulate and dissolved matter: a case study in Tyrrhenian Sea by Durrieu de Madron et al.

**Overview**

The manuscript by Durrieu de Madron et al. presents CTD, ADCP and optical profile data along a section crossing the Tyrrhenian Sea and investigates salt finger staircase effects on particulate and dissolved matter distribution. The authors have significantly improved the readability and presentation of their results, but there are still a few corrections and clarifications to be made, and some points that could be improved. Therefore, I recommend a minor revision based on the comments below.

We thank Reviewer 3 for their constructive feedback and positive evaluation of our revised manuscript. We have addressed the remaining comments and suggestions with care in this new version.

**1 General comments**

a) Add Richardson number Ri to Fig. 3, and buoyancy frequency N2, vertical shear S2, Ri depth profiles in Figs. 6 and 7; expand the discussion on turbulent mixing in terms of shear instability to support your claims on lines 414–443.

We have added to Figure 3 the distribution of Richardson, Ri, values along the section. We have also added to Figures 6 (station 20) and 7 (station 09) the profiles of buoyancy N (in cph), Ri, and a comparison between S2 and  $4\times$ N2 (to delineate the region where Ri

We have verified that the sub-plots in each figure have been labelled correctly.

Line 327: tongues?

We changed the wording to 'tongue-like feature'.

Line 359, 367: delete 'and' Fig. 7 caption: 'Major density steps are delineated by horizontal grey lines. The horizontal dashed lines indicate the base of the main density interfaces.' – these two sentences are redundant

In order to maintain the clarity of figures containing numerous subplots, we have removed the density step boundaries.

Line 373: position

Done

Line 393: (a) Depth

Done

Line 394: basin measured by Fig. 8 is inconsistent between (a) and (b). Firstly, (a) is not marked. I presume the dots in (b) are steps identified using the procedure referenced in Methods; dots are lacking from (a). (a) shows isopycnals, (b) does not. What is 'average layer depth' in y axis label in (b), is it just depth (the float was profiling, not in isopycnal following mode, correct?)? Please present consistent information between the panels and clearly explain what is illustrated in the caption.

Yes, the float was profiling. We have redrawn the figures so that they are comparable (both include now the points indicating the stages identified using the procedure mentioned in the Methods section, as well as the isopycnals). The legend has been completed.

Line 404: wind forcing

Done

**Line 427: of concurrent direct**

We have revised the sentence to remove the term 'direct' as suggested, to improve clarity. The revised sentence now reads: 'While this co-occurrence does not establish causality, and in the absence of turbulence measurements, the potential disruptive effect of this eddy on staircase formation warrants consideration'.

Line 429: delete 'This observation is consistent with the emerging understanding of the dynamic interplay between mesoscale features and fine-scale thermohaline structures in the ocean interior.' – redundant

Done

Line 434: highlighted the persistence of

Done

Line 467: in suspended

Done

Line 468: coarse particle fraction

Done

Line 468–469: 'microscale' in the ocean is O (mm)-few cm! And 'macroscale' is not established nomenclature for a specific vertical scale. Please replace 'macroscale' and 'microscale' by something like 'scales between/of the order of ...' to be specific.

We have revised the sentence to replace the terms 'macroscale' and 'microscale' with more precise descriptions of the vertical scales involved, as suggested. The sentence now reads: 'The overall diminution in suspended particle abundance, concurrent with an augmentation of the coarse particle fraction, is evident not only at large vertical scales throughout the water column beneath the Eastern Intermediate Waters (EIWs), but also at finer scales spanning several decametres within the thermohaline staircases.'

Line 506: delete 'in the literature'

Done

Line 515: interfaces

Done

Line 542: sedimentation patterns, promoting

Done

Line 546: organic matter?

Done

Line 572: delete 'in'

Done

Line 577: assessed at the depth of each step 2

Done

---

## Author Response (AR3)

Title: Effect of double diffusion processes in the deep ocean on the distribution and dynamics of particulate and dissolved matter: a case study in Tyrrhenian Sea

Author(s): Xavier Durrieu de Madron et al.

MS No.: egusphere-2024-3436

MS type: Research article Iteration: revised, submission #3

My main comment on the second revised version concerned the lack of clear evidence of retention—aggregation process. To address this point, the authors have now included two new plots which make use of a larger number of CTD profiles to statistically show the influence of staircases on the Junge index and the AOU. In my view, this is a useful addition that provides clearer evidence of the retention—aggregation process. While the differences between the two groups of profiles (with and without staircases) in the decreasing slopes of the mean Junge index and AOU with depth are weak, they exist. This new figure provides more convincing arguments than the former versions of the manuscript, and I thank the authors for this improvement. Statistically speaking, more could probably be done, but as it is now, the manuscript has already gained in strength, and the arguments are more convincing to readers.

I don't have any further major comments regarding the core of the manuscript.

We thank the reviewer for their positive comments regarding the statistical analysis. We have also carefully considered their remarks on the validity and relevance of calculating the Richardson number.

However, this new version includes a point about the Richardson number (Ri) in response to a query from one of the other two reviewers. Using this Ri causes confusion and does not provide any useful information:

My first comment relates to the way Ri is calculated, which makes use of buoyancy frequency (from CTD) and shear (from LADCP) profiles, both smoothed over a large scale (48 m, l. 268). At this scale, it is unlikely that shear-driven favourable regions will be detected. This is because, at these scales, the shear is usually far too weak and that open-ocean instabilities far from boundaries occur at meter-scale. The only reason Ri is sometimes below the theoretical value of 1/4 in this dataset is due to the very weakly stratified layers. Shear-driven turbulence is driven by shear, not by weak buoyancy!

From my experience, even with a 10-m scale LADCP shear and N2, it is difficult to associate Ri (say lower than 1/4, or lower than some number such as 1 to account for the low vertical resolution of the LADCP shear profiles) with, for instance, enhanced levels of turbulent kinetic energy dissipation rates (i.e. measured turbulence levels) unless you sample very active, shear-driven turbulent regions. Thus, using 48-m scale resolution in an open-ocean staircase environment, a worse configuration, will only exhibit weakly stratified regions, which could be achieved using N2 only. Attempting to link Ri

Line 422: There's a misalignment issue in Figure 5 with the numbers in the subplot.

Thank you for pointing this out. If the comment refers to the relative size of the first subplot compared to the others, this issue has been corrected. However, if the misalignment concerns the color bar values, please note that we were unable to adjust them, as the formatting is automatically managed by Ocean Data View (ODV) based on the number of decimal places, and cannot be manually modified.

Figure 9: When you mention shifted AOU profiles, do you mean both graphs start at 700 m, even though the linear regression for AOU begins at 1000 m? If so, the last two lines of the caption may be more confusing than helpful, as the graph is self-explanatory. Additionally, depth is plotted on the y-axis in all other figures and plots, including those for individual stations ... except in this one. If this does not affect the readability of the plot from a statistical perspective, you might consider flipping the axis.

The AOU shift applied in Figure 9 concerns the AOU values, not the vertical position. This shift was introduced to better visualize and compare the oxygen evolution within the staircase layer across profiles. We have clarified this point in the figure caption to avoid confusion.

As for the axis orientation, we have now reversed the axes so that depth increases downward, consistent with the convention used in all other figures.

Line 612: You define WStairs and NoStair in 2.3, and paragraph 3.2.2 is about the positioning and persistence of the main staircases.

A reference to the WStairs group (profiles with multiple thermohaline staircases) has been added to clarify this ambiguity pointed out by your remark. The distinction between 'Wstairs' and 'NoStair' profiles lies primarily in the presence or absence of well-defined staircase structures in the 500–1000 m layer. The persistence described in section 3.2.2 refers specifically to the deep staircases observed below 1000 m. These clarifications have now been included in the revised text of section 2.3.

Line 614: If I am not mistaken, Aiken et al., 1991 refers to a book. Could you specify the exact location of this reference within your manuscript?

The reference specifically concerns Chapter 2 of the book: Interactions Between Continuous Predictors in Multiple Regression by Aiken, West, and Reno (1991). We have updated the reference in the bibliography to indicate the relevant chapter, rather than citing the entire book.

Lines 610–625: While the slopes are very close in value, you report a p-value indicating that the difference is statistically significant. But, given the small magnitude of these differences and the brief description of your statistical approach, this section could benefit from a more detailed explanation to clarify the reasoning behind the significance and its implications. For instance, you could include some details about the statistical approach in the Methods

section (paragraph 2.3), which would help keep the discussion section more concise and focused without losing information.

While the slope differences are indeed small in magnitude, the statistical tests indicate that the effects are nonetheless detectable. This suggests that the presence of thermohaline staircases has a measurable, albeit moderate, impact on the vertical dynamics of particles, as inferred from parameters such as the Junge index and AOU.

In response to the reviewer's comment, we have added a more detailed description of the statistical methods used in the Materials and Methods section (paragraph 2.3). We have also slightly reworded the discussion text accordingly.

**Third review of EGUsphere-2024-3436**

Effect of double diffusion processes in the deep ocean on the distribution and dynamics of particulate and dissolved matter: a case study in Tyrrhenian Sea by Durrieu de Madron et al.

**Overview**

The manuscript by Durrieu de Madron et al. presents CTD, ADCP and optical profile data along a section crossing the Tyrrhenian Sea and investigates salt finger staircase effects on particulate and dissolved matter distribution.

The authors have adequately addressed all the reviewers' comments which has strengthened the manuscript. I only have some minor further comments below.

We deeply thank the reviewer for its additional constructive comments and suggestions.

Line 308: stratification is lower

Corrected

Figure 3d: Ri would be best shown in log10 scale to emphasize Ri

Line 322: in the upper 300 m

Corrected

Line 428: across the basin from west to east during

**Corrected**

Line 461: absent in the upper 1000 m at the stations within the anticyclonic eddy

Corrected

Line 473: delete 'drops'

Corrected

Line 613: particulate organic matter?

Indeed. Corrected

Line 653: vertical flux

Corrected

Line 667: vertical fluxes

Corrected

---

## Author Response (AR4)

Line numbers are from the tracked-changes version line 761: sigma\_theta is the potential density anomaly (not the specific volume anomaly).

Corrected.

Strictly, PSU is not accepted (but it is still seen in some papers). Practical Salinity (unitless) is given in the practical salinity scale. In the introduction, Li 47 gives the salinity gradient in PSU m-1, whereas line 66 refers to literature with Absolute Salinity (in units of g kg-1). The latter is the standard we encourage; however, the use of Practical Salinity is accepted as long as this is stated clearly. Because the gradient will be the same (and you are using an approximate sign), I suggest you replace 0.5 PSU m-1 to 0.5 g kg-1 m-1. Or specify as "salinity measured in practical salinity scale (0.5 m-1)".

Thank you for pointing this out. It is true that we could have used absolute salinity and conservative temperature from the outset of this study, but for reasons of characterizing water masses and comparing with the literature, we preferred to stick with the practical EOS-80 scales. We have made the necessary corrections to remove the reference to PSU.

Line 156: replace salinity with practical salinity.

Corrected.

In line 158, insert, "The practical salinity is measured in the practical salinity scale."

Corrected.

Li 185: for practical salinity

Corrected.

Li 210, 412: consider removing PSU

Corrected.

Ideally, we do not want PSU in the figures, but because the required changes are substantial, I leave this up to you.

All the relevant figures have been corrected.

Where you use sigma-1, in figure captions, please specify as "potential density anomaly, \sigma\_1 referenced to 1000 dbar pressure".

Corrected.

Figure 13 seems like sigma\_0. Please specify as "potential density anomaly, \sigma\_0 referenced to the surface pressure".

Corrected.